# An estuarine tuned Quasi-Analytical Algorithm (QAA-V): assessment and application to satellite estimates of SPM in Galveston Bay following Hurricane Harvey

Ishan D. Joshi[1], Eurico J. D'Sa[1,*]

[1]Department of Oceanography and Coastal Sciences, Louisiana State University, Baton Rouge, LA 70803, USA; (ijoshi1@lsu.edu, ejdsa@lsu.edu)

*Correspondence to*: Eurico J. D'Sa (ejdsa@lsu.edu; Tel: +1-225-578-0212)

**Abstract.** The standard quasi-analytical algorithm (Lee et al., 2002) was tuned as QAA-V using a suite of
synthetic data and in-situ measurements to improve its performance in optically complex and shallow estuarine waters. Two modifications were applied to the standard QAA: 1) a semi-analytical relationship for obtaining remote sensing reflectance just below the water-surface as a function of absorption and backscattering coefficients was updated using Hydrolight® simulations, and 2) an empirical model of the total non-water absorption coefficient was proposed using a ratio of green to red bands of an ocean color sensor, which is known
to work well in various inland and estuarine environments. The QAA-V derived total absorption and backscattering coefficients, which were evaluated in a variety of waters ranging from the highly absorbing and turbid to relatively clear shelf waters, showed satisfactory performance on Hydrolight®-simulated synthetic dataset ($R^2$>0.87, MRE<17 %), in-situ estuarine & near-shore dataset ($R^2$>0.70, MRE<35 %), and the NOMAD dataset ($R^2$>0.90, MRE<30 %), respectively. When compared to the standard QAA (QAA-v6), the QAA-V
showed an obvious improvement with ~30–40 % reduction in absolute mean relative error for Hydrolight®-simulated synthetic and in-situ estuarine & near-shore datasets, respectively. . The methodology of tuning QAA was applied to the VIIRS ocean color sensor and validation results suggest that the proposed methodology can also be applied to other ocean color & land-observing sensors. The QAA-V was also assessed on VIIRS imagery using a regional relationship between suspended particulate matter (SPM) and particulate backscattering
coefficient at 532nm ($b_{btnw}532$) ($R^2$=0.89, N=33). As a case-study, the QAA-V processing chain and VIIRS imagery were used to generate a sequence of SPM maps of Galveston Bay, Texas following the unprecedented flooding of Houston and surrounding regions due to Hurricane Harvey in August 2017. The record discharge of floodwaters through two major rivers into the bay resulted in very high SPM concentrations over several days throughout the bay, with wind forcing additionally influencing its distribution into the coastal waters of the

northern Gulf of Mexico. The promising results of this study suggest that the application of QAA-V to various ocean color and land-observing satellite imagery could be used to assess the bio-optical state and water quality dynamics in a variety of coastal systems around the world.

[**Keywords**: QAA, VIIRS, SPM, Hurricane Harvey, Galveston Bay, ocean color]

## 5   1 Introduction

Urbanization and associated anthropogenic stressors are of major concern for ecosystem health and water quality of estuarine environments, affecting cumulatively the coastal and marine ecosystems through estuarine-shelf exchange processes (Haynes et al., 2007; Bricker et al., 2008; Jutterström et al., 2014). Inherent optical properties (IOPs) such as absorption and backscattering coefficients have immense potential to capture changes in the bio-optical state of an aquatic system and hence, provide crucial information about regional alterations in water quality associated with terrestrial pollution (Zielinski et al., 2009), harmful algal bloom (Hu et al., 2008), floods (Álvarez-Romero et al., 2013), hurricanes (Lohrenz et al., 2008; Chen et al., 2009), seasonal cycles (D'Sa and Miller, 2003; D'Sa et al., 2006; Singh et al., 2010; Joshi and D'Sa, 2015), and even man-made catastrophe such as oil spills (Ramsey III et al., 2011; D'Sa et al., 2016). In addition to water itself, there are three major water constituents that contribute to the water-leaving radiance ($L_w$), namely, colored dissolved organic matter (CDOM; also called as "gelbstoff" or "*gilvin*"), suspended sediments (detritus and minerals), and phytoplankton. CDOM and suspended sediments are strongly associated with light absorption in the blue and, therefore, high concentrations may reduce light quality for the photosynthetic organisms (*e.g.*, phytoplankton and submerged vegetation) in estuarine waters (Keith et al., 2002; Ralph et al., 2007; Pedersen et al., 2012). The effects of water turbidity caused by dissolved and particulate components on the physical and behavioral changes of aquatic species have been well reported in the literature (Wang et al., 2008; Kjelland et al., 2015). Collectively, these water constituents attenuate incoming light, while a fraction of it is backscattered out of water by the water itself and particles. Therefore, deciphering $L_w$ (or remote sensing reflectance-Rrs) to separate the individual contributions of optically active components may provide crucial information about the bio-optical state of a water body.

Field-based sampling methods are traditional and accurate ways to measure bio-optical properties; however, they lack adequate spatial and temporal coverage for capturing short-period bio-optical alterations and estuarine-scale dynamics. In contrast, remote sensing platforms (*e.g.*, satellite sensors) sense $L_w$ (or Rrs) signal and are advantageous over field observations in providing better synoptic spatiotemporal coverage if the signal is successfully linked to in-water IOPs. As such, satellite-based remote sensing has been widely used to monitor

harmful algal blooms (Carvalho et al., 2011; Hu et al., 2016), pollution events (Mishra et al., 2013; Zhao et al., 2014), suspended sediment dynamics (D'Sa et al., 2007), CDOM distribution and carbon flux (Joshi and D'Sa, 2015; Joshi et al., 2017a), phytoplankton biomass and primary production (Uitz et al., 2010; Matsumoto et al., 2014), and to evaluate the effects of climate change on exotic marine biota (Liu et al., 2006; Castillo and Lima, 2010; Cavanaugh et al., 2011; Pu and Bell, 2017).

In recent years, both empirical and semi-analytical models have been frequently used to link satellite observations and in-water properties, such as IOPs, vertical diffuse attenuation coefficients-$K_d$, suspended particulate matter concentrations-SPM, CDOM, pigment concentrations, phytoplankton cell counts and cell size, and particle size (D'Sa et al., 2003; 2006; 2007; Pan et al., 2010; Chen et al., 2013; Brewin et al., 2015; Joshi et al., 2017b). Empirical relationships are mathematical formulations (*e.g.*, simple or multiple regressions) that directly link water-leaving measurements to the parameter of interest in surface waters. They are simple in nature, easy to implement, and do not require deep understanding of the underlying relationships between light and water properties. Because the performance of empirical relationships is uncertain outside the range of observations that are used to develop them, their applicability is doubtful and may cause significant errors if used in waters with different optical properties. In contrast, semi-analytical models, which are based on radiative transfer theory, invert Rrs using a suite of analytical and empirical relationships to derive the water IOPs (absorption and backscattering coefficients of water constituents) (Lee et al., 2002). Because they solely depend on the water-leaving radiance and require less information about in-water bio-optical properties, they have better applicability and accuracy as compared to the empirical methods in a variety of waters (IOCCG, 2006). However, a major drawback is that the retrieval of IOPs for individual water components is strongly dependent on the performance of the respective empirical models.

To some extent, this drawback is minimized in a multiband quasi-analytical algorithm (QAA) for optically deep waters (Lee et al., 2002). This algorithm analytically decomposes total non-water absorption coefficients ($a_{tnw}$) for combined CDOM and suspended sediments ($a_{dg}$) and phytoplankton ($a_\phi$) using their spectral information. The QAA has been improved (e.g., QAA-v5 and QAA-v6) for better performance in the turbid coastal waters (Lee et al., 2009). Several studies have also contributed to the standard QAA with regional/global modifications in a variety of waters such as turbid waters of the Mississippi and Atchafalaya River system (Zhu et al., 2011), coastal waters of South China Sea (Dong et al., 2013), and inland waters of the USA and China (Li et al., 2013). A large number of studies have evaluated the standard QAA in different regions (e.g., turbid inland waters of northeast China, shallow ponds of the northwestern Mississippi, Lake Taihu, Yellow Sea, East China Sea, Arctic, and low-latitude oceans) with acceptable performance in coastal and oceanic waters,

but decreasing accuracy towards CDOM-rich and sediment-rich estuarine and inland waters (Lee et al., 2010; Qing et al., 2011; Zhu et al., 2011; Mishra et al., 2013; Mitchell et al., 2014; Zheng et al., 2014; Pitarch et al., 2016). Several factors could be responsible for the QAA's poor performance in shallow waters, including: 1) the empirical relationships of QAA were designed using field observations in coastal and oceanic environments and may not be suitable for the optically deep estuarine and near-shore waters, and 2) majority of empirical models of the standard QAA use Rrs at blue wavelengths (e.g., 443nm and 490nm); however, it is well-known that satellite products suffer from large errors at short wavelengths due to uncertainties in atmospheric-correction, especially in coastal waters. This suggests the need for an estuarine-specific tuning of the QAA and its evaluation and application to newer ocean color satellite sensors.

In this study, we present a tuned multiband Quasi-Analytical Algorithm (QAA-V) that is optimized primarily for the Visible and Infrared Imaging Radiometric Suite (VIIRS) ocean color sensor and calibrated for various other ocean color sensors, such as Sentinel-3 Ocean and Land Colour Instrument (Sentinel3 OLCI), MODerate resolution Imaging Spectroradiometer (MODIS-Aqua), MEdium Resolution Imaging Spectroradiometer (MERIS) and Sea-viewing Wide Field-of-view Sensor (SeaWiFS), and land-observing sensors, such as Landsat-8 Operational land imager (Landsat8 OLI) and Sentinel2 Multispectral instrument (Sentinel2 MSI) to estimate IOPs in shallow estuarine and near-shore waters. First, synthetic data were generated using Hydrolight® simulations (Mobley and Sundman, 2013) for highly absorbing and scattering waters, and used collectively with estuarine in-situ observations to update coefficients for the semi-analytical and empirical models of the standard QAA processing chain. The algorithm's performance is then evaluated on three datasets: 1) Hydrolight® simulated dataset, 2) a subset of well-known National bio-Optical Marine Algorithm Dataset (NOMAD), and 3) field observations that were obtained in various estuaries in the U.S. East Coast and the Gulf of Mexico. The QAA-V performance was compared to the QAA-v6, which was mainly tuned to improve QAA's performance in turbid coastal waters. Additionally, using a linear backscattering–SPM relationship, the QAA-V's applicability to VIIRS and estuarine waters is analysed for SPM of various coastal sites, including Galveston Bay (USA). Finally, as a case study, VIIRS-derived SPM imagery of Galveston Bay were obtained following Hurricane Harvey to assess SPM dynamics in the bay and its impact on the coastal ocean.

## 2 Materials and methods

### 2.1 Data for QAA-V algorithm

Three datasets were used for tuning and evaluating QAA-V's performance in a variety of waters ranging from highly turbid estuarine environments to the relatively clear shelf waters. These datasets include: 1) a synthetic dataset, 2) NASA's bio-optical marine algorithm dataset, and 3) an estuarine dataset. Availability of numerous observations representing a true state of natural systems is the primary requirement for any algorithm development and validation analysis. Hydrolight® radiative transfer model (Mobley and Sundman, 2013) was used to generate a large set of synthetic data (HL; N=561) for tuning and extending QAA-V's ability to perform in highly absorbing and highly scattering waters (*e.g.*, turbid estuarine environments). The process of generating synthetic data using Hydrolight® simulations was similar to the International Ocean Color Coordinating Group report (IOCCG Report-5) (IOCCG, 2006) and is briefly described in the supplementary section (S1) with necessary modifications based on in-situ estuarine observations. In-situ estuarine & near-shore dataset (IES) included 340 concurrent water inherent optical properties (IOPs; e.g., absorption and backscattering coefficients) and above-water Rrs measurements at various locations in the U.S. East Coast and the northern Gulf of Mexico (Fig. 1a). Data were compiled from NASA's SeaBASS repository by applying a depth threshold of 10 m for obtaining measurements in estuarine and near-shore waters (Werdell et al., 2003). The IES dataset was further divided into a training set (EcoHAB & Tampa Bay; N=121) and a testing set (N=219) for tuning and validating QAA-V, respectively (Table 1). NOMAD (NASA bio-Optical Marine Algorithm Dataset) is a freely available, high quality field dataset for ocean color algorithm development and validation (Werdell and Bailey, 2005). It includes IOPs and Rrs collected in waters ranging from oceanic to estuarine environments, but mostly in shelf waters around the world. We extracted a subset (N=547) containing complete observations of IOPs and Rrs (Fig. 1b). Data distributions of synthetic data clearly showed the representation of CDOM-rich and sediment-rich waters, whereas phytoplankton absorption was of secondary importance as generally observed in several estuarine environments (Fig. 2). The training and testing data (HL and IES datasets) ranged from approximately 0.1 to 7 m$^{-1}$ for the CDOM absorption coefficient ($a_g443$), 0.05 to 4.5 m$^{-1}$ for (detritus+minerals, or non-algal particles) absorption coefficient ($a_{NAP}443$), 0.05 to 2 m$^{-1}$ for the phytoplankton absorption coefficient ($a_\phi443$), and 0.04 to 0.2 m$^{-1}$ for the particle backscattering coefficient ($b_{btnw}532$) (Fig. 2).

## 2.2 Data for Galveston Bay

Galveston Bay, the seventh-largest estuary in the United States (area= ~1600 km$^2$; mean depth= ~2 m), is located along the upper coast of Texas in the northern Gulf of Mexico (Fig. 1c). The Trinity River is the major source of fresh water (~50 %) to the bay followed by the San Jacinto River (~30 %) and local watersheds (~20 %) (Guthrie et al., 2012; Lucena and Lee, 2017). With the busiest petrochemical port in the US, Galveston Bay experiences frequent oil spills; ~3,500 oil spills (~4,16,000 gallons) incidences were reported between 1998-2009 (Lester and Gonzalez, 2011). The bay is connected to the Gulf of Mexico via 3 passes: Bolivar Roads Pass, Rollover Pass, and San Luis Pass. Galveston Bay can be divided into four sections, 1) Trinity Bay (TB), 2) upper Galveston Bay (UGB), 3) the lower Galveston Bay (LGB), and 4) East Bay (EB) (Fig. 1c).

Surface water samples were collected at several stations during two field surveys on September 29, 2017 and October 29–30, 2017 as part of a larger study to investigate the after-effects of Hurricane Harvey (August 25–29, 2017) on the water quality of Galveston Bay. Available measurements of suspended particulate matter (SPM) concentration were utilized for evaluating the applicability of QAA-V in estuarine environments. Samples were filtered using pre-combusted and pre-weighed 47 mm, 0.7-µm porosity Whatman GFF filters for SPM concentrations (Neukermans et al., 2012). An analytical scale with an accuracy of ±0.1 mg was used to measure mass of SPM. Profiles of $b_{btnw}$ were obtained at each station using the WETLabs VSF-3 (470 nm, 530 nm, 670 nm) and ECO BB (532 nm) backscattering sensors (D'Sa et al, 2006) and surface values were averaged for depth <1 m. Above-water measurements of water-surface radiance ($L_w$, Wm$^{-2}$nm$^{-1}$sr$^{-1}$, nadir=40$^o$-50$^o$, azimuth=90$^o$– 135$^o$), sky radiance ($L_{sky}$, Wm$^{-2}$nm$^{-1}$sr$^{-1}$, zenith=40$^o$–50$^o$, azimuth=90$^o$–135$^o$), and reference-plate radiance ($L_{plate}$, Wm$^{-2}$nm$^{-1}$sr$^{-1}$, nadir=0$^o$, azimuth=90$^o$–135$^o$) were collected using GER1500 512iHR spectroradiometer under clear-sky conditions (Mobley, 1999). The spectroradiometer was set to provide an average of 3 internal scans for considering the variability in reference and target conditions. Hence, the final spectrum was an average of 9 spectra (3 replicates with 3 internal scans per measurement) at each station (Joshi et al., 2017a). The above-water remote sensing reflectance (Rrs$^{0+}$, Unit: sr$^{-1}$) was obtained using the following equations (Mueller et al., 2003),

$$Downwelling\ Irradiance\ (E_d)\ (Wm^{-2}nm^{-1}sr^{-1}) = \pi \times \frac{L_{plate}}{\rho_{plate}} \tag{1}$$

$$Above-water\ remote\ sensing\ reflectance\ (R_{rs}^{0+})\ (sr^{-1}) = \frac{L_w - \rho \times L_{sky}}{E_d} - R_{rs}(residual) \tag{2}$$

where $\rho_{plate}$ is reference plate reflectance (99%) and $R_{rs}(residual)$ is attributed to residual sky-radiance which was taken as Rrs$^{0+}$(950 nm) (Mobley, 1999).

In addition, discharge data were acquired from USGS river gauge sites for the Trinity River (Romayor site-USGS 08066500 & Wallisville site-USGS 08067252) and at the San Jacinto River (the eastern flank-USGS 8070200 & the western flank-USGS 8068090) for examining variations in freshwater flows after Hurricane Harvey. Wind speed and direction were acquired from NOAA Eagle Point station (ID-8771013) in Galveston Bay. Level-1 VIIRS (9 images) and MODIS-Aqua (1 image) products were obtained from NASA's Ocean Color data archive (OBPG, NASA), including three images during field surveys in Galveston Bay. VIIRS imagery was not available on September 29, 2017 corresponding to the first field survey; hence, the next day image (September 30, 2017) was used in this analysis. Sentinel3 OLCI Level-2 image was downloaded from Earth Observation Portal (EUMETSAT) and Landsat8 OLI Level-1 image was downloaded from USGS Earth Explorer for October 29, 2017.

### 2.3 QAA processing chain

The underlying structure of QAA-V is similar to the standard quasi-analytical algorithm (QAA; Lee et al., 2002) and its modifications (e.g., QAA-v5 and QAA-v6) suitable to coastal and open oceans (Lee et al., 2007; Lee et al., 2009). The processing pathway of QAA-V is illustrated in Table 2 and briefly mentioned here with justifications for necessary modifications. The QAA and its updated versions rely upon the principle that the spectral remote sensing reflectance just below the water surface ($Rrs^{0-}$) is a function of the spectral backscattering and absorption coefficients (Gordon et al., 1988) and it can be modeled using the following equation (Table 2 – Level 1A),

$$R_{rs}^{0-}(\lambda) = g_0 \times u(\lambda) + g_1 \times [u(\lambda)]^2 \text{ (sr}^{-1}); \; u(\lambda) = \frac{b_{bt}(\lambda)}{a_t(\lambda) + b_{bt}(\lambda)} \tag{3}$$

$a_t$ and $b_{bt}$ are total absorption coefficient and total backscattering coefficient (m$^{-1}$), respectively. $Rrs^{0-}$ can be easily computed from above surface remote sensing reflectance ($Rrs^{0+}$) using the following relationship (Lee et al., 1999) (Table 2 – Level 0),

$$R_{rs}^{0-}(\lambda)(sr^{-1}) = \frac{R_{rs}^{0+}(\lambda)}{(0.52 + 1.7 \times R_{rs}^{0+}(\lambda))} \tag{4}$$

The coefficients $g_0$ and $g_1$ are empirically derived parameters related to the directional nature of the upwelling light field (Q), and f (well-known as f/Q term). These coefficients depend on sun angle, viewing geometry, wind speed, and the bio-optical state of natural waters and vary with phase function (Morel and Gentili, 1991; 1993; 1996). Thus, appropriate coefficients are needed for different aquatic environments (Lee et al., 2002). For example, the values of $g_0$ and $g_1$ were previously suggested as 0.0949 and 0.0794, respectively, for

the oceanic waters (Gordon et al., 1988). Later, better approximations of $g_0$ (0.0895) and $g_1$ (0.1247) were proposed for the reflective coastal waters using radiative transfer models on simulated data (Lee et al., 1999). Average values of $g_0$ and $g_1$ from Gordon et al. (1988) and Lee et al. (1999) were also used for both coastal and oceanic waters (Lee et al., 1999). The synthetic data for obtaining these historic values of $g_0$ and $g_1$ were

generated using Case-1 radiative transfer models. As the first modification, we updated these coefficients using Hydrolight® simulations with a 4-component case-2 model (supplementary S1) because a shallow water environment was the main focus of this study. In addition, it has been previously suggested that the molecular scattering may primarily contribute to the Rrs especially in the blue and green wavelengths in oceanic waters. However, the phase function effect of water molecules could be much smaller than that of particles in near-shore

and estuarine waters. Hence, we have avoided separating Eq. (3) as it is usually done for accounting phase-function effects of individual backscattering contributors (Lee et al., 2013; Zheng et al., 2014).

Next, QAA-v6 uses a set of empirical models (e.g., Eq. 5) based on above-surface Rrs threshold (0.0015 $sr^{-1}$) to estimate total non-water absorption coefficient at a reference wavelength in coastal and oceanic waters (Table 2 – Level 1B),

If $R_{rs}^{0+}(670) < 0.0015$ $sr^{-1}$

$$a_{tnw}(\lambda_0) = 10^{(-1.146-1.366 \times x - 0.469 \times x^2)}, \text{ where } x = \log_{10}\left(\frac{R_{rs}^{0-}(443)+R_{rs}^{0-}(490)}{R_{rs}^{0-}(\lambda_0)+5 \times R_{rs}^{0-}(670) \times \frac{R_{rs}^{0-}(670)}{R_{rs}^{0-}(490)}}\right),$$

$$\text{else, } a_{tnw}(\lambda_1) = 0.39 \times \left[\frac{R_{rs}^{0+}(670)}{R_{rs}^{0+}(443)+R_{rs}^{0+}(490)}\right]^{1.14} \tag{5}$$

where $a_{tnw}$ is total non-water absorption coefficient ($m^{-1}$) and $\lambda_0$ = 555 and $\lambda_1$ = 670 nm.

We avoided using blue wavelengths in our empirical models as blue bands, especially 443 nm or lower

bands suffer from large errors in atmospheric-correction due to the high abundance of CDOM and suspended particles, and absorbing aerosols in a coastal environment. In contrast, the green to red band ratio (GRBR) can be used for estimating the absorption coefficient of an individual water constituent with the primary condition of their dominance in the study region. The GRBR has been used, for example, to monitor water constituents in various estuarine and coastal waters, e.g., $a_g355$ in Barataria Bay, USA (Joshi and D'Sa, 2015), $a_g412$ in

Apalachicola Bay, USA (Joshi et al., 2017a), $a_{dg}412$ in Galveston Bay, USA (D'Sa et al., submitted), suspended particulate matter (D'Sa et al., 2007), and chlorophyll index (harmful algal bloom) in the northern Gulf of Mexico (Qi et al., 2015). Estuarine waters are generally characterized by a high abundance of CDOM, mineral particles, or both and thus, known to have strong light absorption towards shorter wavelengths, sometimes even in the

green region. In contrast, the light absorption in the red region usually remains minimal for CDOM and mineral particles. As a result, the green band can be considered as a pilot band to capture variations in dissolved or mineral particle absorption, whereas the red band as a reference band. Hence, small variations in the GRBR are suitable to capture large variations in absorption at shorter wavelengths due to the exponential nature of CDOM and particle absorption. Similar band ratio (RGCI–red to green chlorophyll index) has been used in a semi-analytical approach to obtain chlorophyll-a concentration for estimating phytoplankton absorption $a_\phi 670$ and subsequently, $a_{tnw}670$ in the productive waters of Tampa Bay, USA (Le et al., 2013). The red to green band ratio (RGBR) works well to quantify variations in chlorophyll concentrations, especially in phytoplankton-dominated waters. Therefore, the GRBR (or RGBR) can be overall associated with the dominant water constituent in the study area. However, optically active water constituents collectively contribute to total light absorption and hence, these band ratios can also be used for the remote estimation of $a_{tnw}$ at green wavelengths, especially in estuarine waters where variations in total absorption coefficients are often noticeable due to high abundance of one or more (e.g., CDOM and particles) water components. A ratio of Rrs at the red and green wavelengths (e.g., 640/645 nm and 555 nm of MODIS) was previously used in the standard QAA to improve the estimates of $a_{tnw}$ at a reference wavelength (e.g., 555 nm for MODIS) in turbid waters, which worked reasonably in coastal waters (Lee et al., 2002; Chen and Zhang, 2015). The processing of high spatial resolution I-1 band (640 nm; spatial resolution: 375 m) is not supported by NASA's SeaDAS tool; therefore, our analysis was limited to the available ocean color bands (M1 to M5; spatial resolution: 750 m).

Additionally, selection of the reference wavelength in Eq. 5 is another important factor affecting the retrieval of IOPs in QAA. The use of a reference wavelength at red wavelengths (e.g., >600 nm) was suggested for the relatively turbid coastal waters where particulate and dissolved absorption is much lower than water absorption (Lee et al., 2002; Aurin and Dierssen, 2012). We used 555 nm as a reference wavelength for three reasons, 1) lack of absorption measurements at red wavelengths (e.g., IES dataset) to tune empirical models (Table 2 – Level 1B), and 2) the use of red reference wavelength would likely deteriorate the estimations in the blue wavelengths due to errors in spectral extrapolation corresponding to the empirical nature of backscattering power law exponent, $\eta$, and 3) a strong relationship between green to red band ratio and total non-water absorption at 555 nm is observed in this study.

Once, $u(\lambda_0)$ is obtained as the positive root of Eq. 3 using $Rrs^{0-}$ and the coefficient $g_0$ and $g_1$, backscattering coefficient at a reference wavelength ($b_{btnw}(\lambda_0)$) can be easily obtained with $u(\lambda_0)$, $a_{tnw}(\lambda_0)$, and the following analytical model (Table 2 – Level 1C0),

$$b_{btnw}(\lambda_0)(m^{-1}) = \left(\frac{u(\lambda_0)}{1-u(\lambda_0)}\right) \times (a_{tnw}(\lambda_0) + a_w(\lambda_0)) - b_{bw}(\lambda_0) \tag{6}$$

where, $a_w$ and $b_{bw}$ are water absorption and backscattering coefficients, respectively. The spectral distribution of particulate backscattering coefficient ($b_{btnw}$) can be modeled using the power-law model (Lee et al., 2002) (Table 2 – Level 1C2),

$$\text{5} \quad b_{bt}(\lambda)\ (m^{-1}) = b_{bw}(\lambda) + b_{btnw}(\lambda_0) \times \left(\frac{\lambda_0}{\lambda}\right)^{\eta} \tag{7}$$

where $\eta$ is the spectral shape of $b_{btnw}$ distribution. The standard QAA-v6 uses the following empirical model to obtain $\eta$,

$$\eta = 2 \times \left(1 - 1.2 \times e^{\left(-0.9 \times \frac{Rrs^{0-}(443)}{Rrs^{0-}(555)}\right)}\right) \tag{8}$$

The variation of power law exponent $\eta$ depends on water properties and size of particles according to
Mie theory and it is extremely difficult to retrieve $\eta$ from Rrs in near-shore waters (Aurin and Dierssen, 2012). Thus, we adopted a different approach of obtaining $\eta$ from $b_{btnw}555$ with a linear relationship that was formulated using field observations in the turbid waters near the Mississippi River's delta (D'Sa et al., 2007) (Table 2 – Level 1C1),

$$\eta = -0.566 - 1.395 \times log_{10}(b_{btnw}555) \tag{9}$$

The spectral distribution of total absorption coefficients was then obtained using $b_{btnw}$ and u (Table 2 – Level 1C3),

$$a_t(\lambda)(m^{-1}) = \left(\frac{1-u(\lambda)}{u(\lambda)}\right) \times b_{bt}(\lambda) \tag{10}$$

To extend and to evaluate the applicability of estuarine-specific QAA tuning, it was further applied to various ocean color (Sentinel3 OLCI, MODIS-Aqua, MERIS, and SeaWiFS) and land-observing sensors
(Landsat8 OLI and Sentinel2 MSI). The calibration coefficients for obtaining total non-water absorption coefficient at a reference wavelength ($a_{tnw}$ ($\lambda_0$); Level 1B in Table 2) are given in Table 3.

## 2.4 Atmospheric-correction of satellite imagery

Level-1 satellite imagery were corrected for the atmosphere using SeaDAS 7.4 image processing tool as described previously (Joshi et al., 2017a) and mean values of a 3x3 pixel box centered at a station location were
considered as reasonable satellite matchups for the field measurements. In addition, in-situ Rrs were matched to the central wavelengths of spectral bands using spectral response functions of respective satellite sensors prior to the sensor-specific tuning and validation,

$$R_{rs_{RSR}}(sr^{-1}) = \frac{\int_{\lambda_1}^{\lambda_2} RSR(\lambda) \times R_{rs_{insitu}}(\lambda)\, d\lambda}{\int_{\lambda_1}^{\lambda_2} RSR(\lambda)\, d\lambda} \tag{11}$$

where RSR = relative spectral response for satellite sensor, $\lambda_1$ is the lower bound of a spectral band and $\lambda_2$ is the upper bound of a spectral band.

An iterative NIR atmospheric correction scheme was previously evaluated for estuarine environments (Bailey et al., 2010; Werdell et al., 2010; Joshi et al., 2017a); however, it yielded negative Rrs at blue wavelengths and atmospheric-correction failure at several pixels in Galveston Bay. In comparison, the errors in atmospheric-correction were considerably reduced with the MUMM NIR correction as it was designed for low to moderate turbid waters (Ruddick et al., 2006; Novoa et al., 2017). Furthermore, the validation of atmospheric-corrected VIIRS imagery showed reasonable performance of the MUMM atmospheric-correction scheme during both field campaigns in Galveston Bay (Table 4). The MRE, which was relatively higher at the blue wavelengths, was greatly reduced towards the green and red wavelengths. Thus, the success of the atmospheric correction procedure was decided based on the green and red bands, since only these bands were used to tune QAA in this study. However, the observed large errors at other wavelengths (e.g., blue bands) were likely due to the high abundance of CDOM and particles in the study region, and the aerosol model selection in the atmospheric correction procedure (Minu et al., 2014). The time-difference between field and satellite measurements resulted in an error enhancement at longer wavelengths with relatively smaller errors in October (difference= 0 day) and larger errors in September (difference= +1 day) (Table 4). Overall, low MRE in the green and red wavelengths indicated the usefulness of the MUMM atmospheric-correction for investigating the bio-optical properties with QAA-V processing chain and VIIRS ocean color data in Galveston Bay (USA).

## 2.5 Statistical analysis

The algorithm's performance and atmospheric-corrected VIIRS imagery were evaluated using coefficients of determination ($R^2$), root mean square error ($RMSE_{log10}$), bias ($Bias_{log10}$), and absolute mean relative error (MRE),

$$Bias_{log10} = \frac{1}{n} \times \sum_{i=1}^{n}[\log_{10}(y_i) - \log_{10}(x_i)] \tag{12}$$

$$RMSE_{log10} = \sqrt{\frac{1}{n} \times \sum_{i=1}^{n}[\log_{10}(y_i) - \log_{10}(x_i)]^2} \tag{13}$$

$$MRE\,(\%) = \frac{100}{n} \times \sum_{i=1}^{n}\left[\left|\frac{y_i - x_i}{x_i}\right|\right] \tag{14}$$

R software was used to generate synthetic IOPs for Hydrolight® case-2 model and the statistical analysis presented in this study.

## 3 Results

### 3.1 Modifications to standard QAA algorithm

A new set of $g_0$ (0.0788) and $g_1$ (0.2379) obtained using Hydrolight® simulated IOPs and Rrs is proposed for the highly attenuating waters of this study. These coefficients showed a notable departure of the quadratic relationship, especially in highly scattering waters when compared to Gordon et al. (1988) and Lee et al. (1999; 2002) (Fig. 3a). We used the historical values $g_0$ (0.0895) and $g_1$ (0.1245) for less reflecting near-shore and shelf waters (i.e., threshold $\rho \geq 0.25$) in the validation analysis (Fig. 3b). Overall, the threshold-based selection of u vs.

$Rrs^{0-}$ model showed a valid retrieval of u ($\approx b_{bt}/(a_t+b_{bt})$) as seen when u was analytically used to obtain total absorption coefficient ($a_t555$) and total backscattering coefficients ($b_{bt}555$) (HL dataset) and vice versa (Fig. 4a & 4b; Table 2 – Level 1A).

    The performance of QAA-V was largely dependent on the estimation of total non-water absorption coefficient ($a_{tnw}$) in level 1B (Table 2). In this study, the empirical model of standard QAA-v6 (Eq. 5) was

replaced by a tuned empirical power-law relationship using a training set (IES; N=121; Table 1) of in-situ observations and Hydrolight® synthetic data (HL; N=561),

$$a_{tnw}(\lambda_0) = \begin{Bmatrix} 10^{(0.139-1.788\times\rho+0.490\times\rho^2} \text{ if } \rho < 0.25 \\ 10^{(0.406-2.940\times\rho+0.928\times\rho^2} \text{ if } \rho \geq 0.25 \end{Bmatrix}; \ \rho = \log_{10}\left(\frac{R_{rs}^{0-}(\lambda_0)}{R_{rs}^{0-}(\lambda_1)}\right) \tag{15}$$

where $a_{tnw}$ is total non-water absorption coefficient and $\lambda_0$ =551 or 555 nm and $\lambda_1$=671 nm.

    A threshold value of 0.25 was set for $\rho$ to merge HL and IES datasets excluding the NOMAD dataset

(Fig. 3b). The $a_{tnw}555$ ($\rho$>1.0) nearly reached the lower limit close to zero for oceanic waters likely due to low concentrations of reflecting and absorbing materials. Thus, we suggest the upper threshold of $\rho$=0.65 beyond which level 1B fails and overestimates water $a_{tnw}(\lambda_0)$. Likewise, negative $\rho$ values of synthetic data represented CDOM-rich waters with very strong absorption even at the green wavelengths (e.g., 555 nm). For HL data, the modeled $a_{tnw}555$ showed a reasonable performance of the green to red band ratio model (MRE=16.3 %,

$bias_{log10}$=−0.0208, $RMSE_{log10}$=0.0963 m$^{-1}$, N=561) (Fig. 4c). The performance of the empirical model, however, showed a significant difference within the IES training data ($R^2$=0.90, MRE=21 %, $bias_{log10}$=−0.0015,

RMSE$_{\log 10}$=0.12 m$^{-1}$, N=120) and testing data (R$^2$=0.72, MRE=34 %, bias$_{\log 10}$ =–0.0294, RMSE$_{\log 10}$=0.19 m$^{-1}$, N=209) (Fig. 4d).

### 3.2 Comparison of QAA-V with the standard QAA-v6

The QAA-v6 algorithm was applied separately on HL synthetic data and IES field observations for a direct
comparison with QAA-V in retrieving optical properties in estuarine waters. Figure 4e shows the performance of QAA-v6 in estimating total non-water absorption coefficient at 555 nm (a$_{tnw}$555) with HL synthetic data. In comparison to QAA-V, the QAA-v6 showed an obvious underestimation at 555 nm especially for a$_{tnw}$555>~0.3 m$^{-1}$. Statistical assessment showed that QAA-V is more accurate than the QAA-v6 with approximately 83 % less bias, 35 % decreased RMSE, and 1.5-fold lower MRE (Fig. 5, Supplementary S2). For IES data, the QAA-v6
showed a clear difference with poor performance at several stations (Fig. 4f) as compared to QAA-V (Fig. 4d). Furthermore, the retrieval errors were large towards upper and lower ends (e.g. a$_{tnw}$555 >~0.3 m$^{-1}$ and <~0.1 m$^{-1}$). Overall, the QAA-v6 had an obvious underestimation with approximately 75 % greater bias, 31 % increased RMSE, and 2-fold higher MRE than the QAA-V at 555 nm (Fig. 5, Supplementary S2).

### 3.3 Evaluation of QAA-V on synthetic HL data, NOMAD data, and IES data

Hydrolight$^®$ simulated case-2 water Rrs$^{0+}$ spectra were fed into QAA-V to derive a$_{tnw}$ and b$_{btnw}$ at 411 nm, 443 nm, 489 nm, and 555 nm (Fig. 6). Although a negative bias indicated overall underestimation of modeled absorption coefficient at 555 nm (Fig. 4c), QAA-V performed satisfactorily in the blue region (Fig. 6a). Furthermore, the modeled a$_{tnw}$ at the blue wavelengths showed relatively lower MRE<13 % and RMSE$_{\log 10}$<0.075 m$^{-1}$ despite the errors in modeled a$_{tnw}$555 at values less than 0.3 m$^{-1}$ (relatively low CDOM and
mineral particle abundance), indicating the secondary importance of the green reference wavelengths in transferring errors to the blue wavelengths in QAA-V's processing chain as compared to the red reference wavelengths (Aurin and Dierssen, 2012). Similarly, QAA-V also estimated b$_{btnw}$ with MRE<16 % at four wavelengths (Fig. 6b). When compared to the QAA-v6, QAA-V performed with approximately 80–90 % lower bias, 30-40 % reduced RMSE, and 1–2-fold decreased MRE at blue wavelengths (Fig. 5, Supplementary S2).
QAA-V's performance on standard NOMAD dataset showed that the MRE was <30 % for all blue wavelengths with a$_{tnw}$443 being the best-retrieved parameter (R$^2$=0.94, MRE=24.3 %, N=547). However, large errors were observed for a$_{tnw}$<0.1m$^{-1}$ at all wavelengths (Fig. 7a).

For field validation, Rrs$^{0+}$ spectra from 15 experiments of IES data were fed to QAA-V for obtaining a$_{tnw}$443 and b$_{btnw}$532 (Fig. 7b and 7c; Table 1). Modeled IOPs showed overall good results for a$_{tnw}$443 (MRE=15

% for training data and 23 % for testing data; Fig. 7b) and $b_{btnw}532$ (MRE=32 % for all data and 26 % for all data except Apalachicola Bay; Fig. 7c). A further analysis of individual datasets showed noticeable variations in MRE. For example, MRE varied from ~11–24 % among five best performing datasets for estimating $a_{tnw}443$, whereas it varied from ~40–75 % for three poorly performing datasets with preliminary processing status (Table 1). On the IES data, QAA-V showed fewer errors in retrieving water IOPs than QAA-v6 (Fig. 5, Supplementary S2).

### 3.4 SPM–$b_{btnw}$532 relationship and validation

A linear relationship ($R^2$=0.89; N=33) was observed between $b_{btnw}532$ and SPM (Eq. 16; Fig. 7d). It was used for evaluating the applicability of QAA-V to get synoptic maps of SPM using VIIRS ocean color sensor in estuarine environments.

$$SPM = 103.07 \times (b_{btnw}532) + 0.24 \; ; \text{where } b_{btnw}532 = \text{backscattering coefficient at 532 nm} \qquad (16)$$

It is important to note that the backscattering sensor had an upper threshold of 0.23 m$^{-1}$ where the sensor saturated and failed to detect variability in SPM; however, the linearity in SPM–$b_{btnw}532$ relationship was assumed beyond the threshold (0.23 m$^{-1}$) in this study. The SPM–$b_{btnw}532$ relationship was compared to a similar relationship developed in turbid coastal waters of the Mississippi River (D'Sa et al., 2007) and a generic multisensor algorithm (Nechad et al., 2010) using IES dataset (Fig. 7e; Table 2). The three algorithms showed quite similar trends despite different water properties of the IES dataset. When compared to field observations of SPM in Apalachicola Bay, Barataria Bay, and Galveston Bay, these algorithms showed good performance with the lowest error for the single wavelength generic algorithm of Nechad et al. (MRE=32.1 %, N=57) and the highest error for SPM–$b_{btnw}555$ relationship of D'Sa et al. (MRE=38.8 %, N=57) (Fig. 7e).

### 3.5 Evaluation of QAA-V in a turbid estuarine environment

The SPM–$b_{btnw}532$ relationship of this study was applied to VIIRS imagery for evaluating satellite-based application of QAA-V in a turbid estuarine environment (Fig. 8). The regional SPM–$b_{btnw}532$ relationship showed ~35 % overall MRE in satellite-field comparison (Fig. 8a) for Galveston Bay during the two field campaigns (Figure 8b-d). Both field and estimated SPM concentrations followed a similar pattern of high to low values along the north-to-south transect. A large-to-small error trend from September 30 (MRE=39.9 %) to October 30 (MRE=26.6 %) was similar to the error trend in the atmospheric-correction of the VIIRS imagery (Table 4). Based on these results, a sequence of cloud-free and atmospherically-corrected VIIRS imagery were

converted to SPM maps using the QAA-V processing chain (Table 2) for analyzing post-hurricane SPM dynamics in Galveston Bay (Fig. 9).

### 3.6 Extending the QAA-V tuning to various satellite sensors

The estuarine-specific green to red band tuning was further applied to evaluate and to extend its applicability to past and present ocean color (e.g., SeaWiFS, MERIS, MODIS-Aqua, and Sentinel3 OLCI) and land-observing sensors (Landsat8 OLI and Sentinel2 MSI) (Table 3). The validation analysis showed promising performance of QAA tuning in obtaining total non-water absorption coefficient ($a_{tnw}443$) and total-non water backscattering coefficient ($b_{btnw}470$) in optically complex and shallow waters of Galveston Bay (Fig. 10). Overall, different satellite sensors showed similar trends of $a_{tnw}443$ and $b_{btnw}470$ along the transect despite having different spectral and spatial sensor resolutions (Fig. 10I – 10IV). The MREs were ~15 %, 9 %, and 12 % for $a_{tnw}443$ retrievals from VIIRS, MODIS-A, and Sentinel3 OLCI sensors, respectively (Fig. 10a-c & 10I), whereas they were ~26 %, 7 %, 22 % for $b_{btnw}470$ retrievals on October 29, 2017 (Fig. 10f-h & 10III). For Landsat8 OLI, these MRE were ~20 % and ~10 % for $a_{tnw}443$ and $b_{btnw}470$, respectively on September 29, 2017 (Fig. 10e, 10j, 10II, & 10IV).

## 4. Discussion

### 4.1 QAA-V algorithm

The standard QAA (Lee et al., 2002) was first tuned with VIIRS sensor bands as QAA-V and then extended to other satellite sensors to obtain more accurate estimates of water IOPs (e.g., total absorption and backscattering coefficients) in shallow estuarine and near-shore waters. For this purpose, two modifications were applied to the QAA, 1) the coefficients $g_0$ and $g_1$ of a quadratic model were updated using Hydrolight® simulations for highly absorbing and highly scattering waters (Eq. 3, Fig. 3a), and 2) an empirical model of QAA for obtaining total absorption coefficient at a reference wavelength was replaced by a set of empirical models that were optimized for highly attenuating estuarine and near-shore waters (Eq. 15, Fig. 3b). The validity of these modifications is demonstrated in Figure 4. The updated coefficients, $g_0$ and $g_1$, showed a significant departure from historical values (Gordon et al., 1998; Lee et al., 1999; 2002) especially at green and red wavelengths in highly attenuating waters. Likewise, these coefficients were updated using a synthetic dataset representing highly attenuating waters and they may not perform satisfactorily in less reflective near-shore and coastal environments. This limitation was addressed by adopting a water-type switching of u vs. $Rrs^{0-}$ algorithm based on a green to red band ratio threshold (ρ) in the QAA-V processing chain (Table 2). The effectiveness of tuning $g_0$ and $g_1$ is also supported by

a previous study showing an ~5-fold bias reduction in obtaining total non-water IOPs at 440 nm (Aurin and Dierssen, 2012).

Overall, negative bias for HL datasets indicated that the threshold-based empirical models underestimated the $a_{tnw}555$; however, a major error in the model performance was observed at the lower end of absorption of values, likely due to two reasons, 1) the failure of the threshold ($\rho$) in providing a smooth switching of empirical relationships between the highly absorbing and scattering waters (synthetic HL dataset) and estuarine and near-shore waters (IES dataset) (Fig. 4b), and 2) the unsuitability of green to red band ratio based empirical relationship for some Rrs spectra in the HL synthetic data (e.g. $a_{tnw}555<0.3$ m$^{-1}$) (Fig. 4a). In contrast, both empirical relationships worked reasonably well in providing a smooth transition from near-shore waters to highly turbid and highly absorbing waters of the IES data (Fig. 4b). However, the large differences within an IES dataset (e.g., training data and test data) could be explained by the processing status of different individual experiments in the IES dataset (Table 1). Training data included Rrs measurements in the final processing status from two well-known experiments, namely, EcoHAB (N=74) and Tampa Bay monitoring program (N=47) (Table 1). Individual absolute mean relative errors (MRE) for these data are 24 % and 16 %, respectively. In contrast, few datasets used in the testing set were in the preliminary processing stage with MRE>35 % (e.g., Chesapeake Bay–Light house, Horn Island, Cojet 7, and Lake Erie).

The validation of QAA-V in a variety of waters yielded reasonable performance as shown in Figures 6 and 7. For HL dataset, an error difference (~2 %) between $a_{tnw}$ (Fig. 6a) and $b_{btnw}$ (Fig. 6b) could be associated with errors in levels 0 and 1A of the QAA processing chain (Table 2). The effect of this systematic error (~1–2 %) is also present in the estimates of backscattering coefficients at blue wavelengths; this could be due to uncertainties in various parameters of the semi-analytical power law model such as the power-law exponent $\eta$ (Table 2; Level 1C1). It has previously been shown that $\eta$ is important for obtaining $b_{bt}$ and its tuning can be responsible for an ~4-fold decrease in the percentage difference for $b_{bt}440$ in estuarine waters than QAA's $\eta$ model (Eq. 8) (Aurin and Dierssen, 2012). As we used a linear model based on $b_{btnw}(\lambda_0)$ for $\eta$ (D'Sa et al., 2007), the modeled backscattering coefficient may suffer from the errors due to uncertainties in the empirical estimation of power law exponent. $\eta$ does not play any role in retrieving $b_{btnw}(\lambda_0)$ and hence, errors associated with $\eta$ should not affect modeled $b_{btnw}(\lambda_0)$. However, the errors in $\eta$ may affect the retrieval of $b_{btnw}$ at other wavelengths (e.g., $b_{btnw}470$) (Table 2). For the NOMAD dataset, the observed large MRE in relatively clearer waters ($a_{tnw}<0.1$ m$^{-1}$) can be due to the fact that the empirical relationships for modeling $a_{tnw}(\lambda_0)$ was not designed to work with very low values of non-water absorption coefficients, as generally observed in the shelf and oceanic environments (Fig. 7a; Eq. 5). For IES datasets, the retrieval of $a_{tnw}443$ is obviously improved compared to $a_{tnw}555$ (Figs. 4d

and 7b); however, 1 to 2-fold increase in errors is observed in IOP retrieval when QAA-V is applied to the field data in contrast to synthetic data. This error enhancement can be due to several factors including, 1) failure to achieve a smooth switching between highly absorbing and reflecting waters and estuarine and near-shore waters in real data (Fig. 4a), 2) inclusion of preliminary data in validation analysis (Table 1), 3) the uncertainty propagation of the previous levels of the QAA-V processing chain (e.g., errors in various empirical, semi-analytical, and analytical relationships) (Table 2), and 4) the difference in methodology of data collection, raw data processing, instrumentation, and measurement and instrument errors. For example, reflectance measurements, the primary input of QAA-V, can have large errors if measured in turbid waters and with large viewing angles (Lee et al., 1999), while bottom contamination may introduce an additional error since in-situ data represent shallow water environments. The bottom effect could probably be reduced with sophisticated correction schemes (Lee et al., 1999; Li et al., 2017; Lee et al., 1998).

A statistical comparison showed that the QAA-V provided better estimations of IOPs than the standard QAA-v6 in estuarine and near-shore waters (Fig. 5). Also, a green to red band ratio is better suited for satellite application of QAA in shallow waters due to fewer errors in these bands (Table 4). In contrast, several studies have demonstrated an over-estimation in blue bands in turbid waters, likely due to errors in the atmospheric-correction (Joshi et al., 2017a; Chen and Zhang, 2015 and reference therein). This over-estimation could lead to the observed underestimation of $a_{tnw}555$ and subsequently, the underestimation of $a$ and $b_b$ at different wavelengths in lower levels of the QAA processing chain (Fig. 5, Table 2).

## 4.2 The SPM optical model

Traditionally, suspended particulate matter (SPM) has been retrieved from remote sensing imagery using single or multiband empirical relationships between above-surface Rrs and SPM concentrations (Doxaran et al., 2002; Miller & McKee, 2004; D'Sa et al., 2007; Han et al., 2016). While empirical relationships are easy to implement, they are regionally limited and may cause large errors if applied to different waters due to differences in particle properties such as absorption, particle size and composition, and refractive index. Furthermore, above-surface Rrs, even at red and NIR wavelengths where particle attenuation controls the Rrs signal, is not a "true" representation of particles as Rrs signal is also marginally contributed by absorption of water and other optically active materials. Hence, this approach has limitation as it can have robust performance only in particle-dominated waters. Another way for estimating SPM concentration is to relate to particle backscattering coefficients ($b_{bp}$). Several studies have reported efficacy of this approach in estimating concentrations of total suspended material

(TSM) and particulate organic carbon (POC) in a variety of waters ranging from estuarine to open oceans (Stramski et al., 1999; Loisel et al., 2001; Aurin and Dierssen, 2012).

When the SPM–$b_{btnw}$532 relationship was applied to the VIIRS imagery, both field and satellite-estimated SPM concentrations showed a similar pattern of high to low along a north-to-south transect in Galveston Bay; however, the differences remained elevated in the turbid region of Trinity Bay (st-2 to st-6 in Fig. 8b and st-16 to st-23 in Fig. 8c). Factors contributing to these differences include, 1) the error propagation from various steps of the QAA-V processing chain to $b_{btnw}$532 (e.g., 20–30 %, Fig. 7c) and hence, further down to the SPM inversion, 2) the uncertainty in the atmospheric-corrected green and red Rrs (e.g., 5–20 %, Table 4), 3) the uncertainty in SPM–$b_{btnw}$532 relationship due to limited observations, 4) the assumption of linearity in SPM–$b_{btnw}$532 model beyond the instrument threshold which may not hold well because $b_{btnw}$532 to SPM ratio depends on the particle nature and it may not always be constant especially in highly turbid waters, and 5) errors in SPM measurements. For example, SPM is usually measured with 0.7-µm (average pore size) GF/F filter, which only represents the total concentration of particles greater than this size. However, while smaller particles may not have significant contributions to the mass-specific property of SPM (e.g., concentration), particles smaller than this size can contribute notably to the underestimation of SPM concentrations in highly turbid waters (Supplementary S3). Furthermore, small particles and even colloidal particles (<0.2 µm) are known to contribute significantly to the total particle backscattering in coastal waters (Zhang and Gray, 2015; Zhang et al., 2011). For instance, high winds associated with the passage of a cold front on October 28, 2017 (not shown) could have resulted in resuspension of smaller particles at the shallower stations on October 29 (Fig. 8c) and contributed to greater differences between satellite estimates and field SPM measurements. Interestingly, on the following day (October 30) under calmer conditions, the differences reduced substantially (Fig. 8d). Thus, the mismatch between mass-specific and optical properties could be a major source of error in the SPM–$b_{btnw}$532 relationship and hence, the observed difference in field-satellite matchups.

### 4.3 Satellite application of QAA-V to Galveston Bay: post-Hurricane Harvey SPM dynamics

A sequence of SPM maps derived from VIIRS imagery using QAA-V revealed interesting patterns of SPM concentrations in Galveston Bay following Hurricane Harvey (Figs. 8 & 9). Hurricane Harvey, a category 4 hurricane, caused catastrophic flooding in the Houston metropolitan area and surrounding regions of Galveston's Bay drainage basin with great potential to degrade the bay's water quality.

The dramatic increase in the discharge of water from the Trinity and San Jacinto Rivers into Galveston Bay (Fig. 9a) following the record rainfall and flooding associated with Hurricane Harvey reveals different patterns of discharge, with the San Jacinto River peaking and retreating to pre-hurricane levels much quicker than the Trinity River. This reflects the differences of the two river basins, as well as the intensity and variability of the precipitation associated with the hurricane in the two basins. A sequence of post-hurricane SPM maps of Galveston Bay (Fig. 9b-9g) reveals distinct spatial and temporal patterns of SPM variations within and outside the bay including the offshore shelf waters that appeared to be strongly influenced by the river discharge and wind forcing. The immediate effect of Hurricane Harvey was clearly evident on August 31 when unusually high SPM concentrations ($>75$ mg L$^{-1}$) were observed throughout the bay corresponding to high freshwater inputs from the Trinity and the San Jacinto Rivers into Galveston Bay (Fig. 9a & 9b). The SPM-rich plume ($>75$ mg L$^{-1}$) extended from the Bolivar Roads pass to a large region of the coastal and shelf waters. In contrast, the bay experienced elevated SPM concentrations on September 02, but the plume was limited by wind forcing and a reduction in freshwater to the bay (Fig. 9a & 9c).

The strong northeasterly winds (~5–6 m s$^{-1}$) observed on September 07 and 08 appeared to restrict the SPM plume closer to the bay entrance and the inner shelf waters (Fig. 9d & 9e). Despite the reduced freshwater inflow into the upper Galveston Bay, sustained fresh water inputs from the Trinity River (~500 m$^3$ s$^{-1}$) and strong northeasterly winds (e.g. sediment resuspension) could have resulted in elevated SPM concentrations throughout the bay. Furthermore, wind-induced downwelling currents appeared to transport low SPM offshore waters near-shore while the high SPM near-shore and plume waters were likely downwelled and could have eventually settled into the shelf sediments.

Despite the noticeable reduction in the Trinity River flow, SPM concentrations remained high within the bay on September 12, indicating the importance of wind-induced sediment resuspension on SPM dynamics in the shallow water environments. However, outside the bay, a well-defined and elevated SPM plume extended offshore, likely associated with the southwesterly winds which induced offshore transport of the inner shelf waters (Fig. 9f). On September 16, total fresh water inflow to Galveston Bay was reduced significantly (~500 m$^3$ s$^{-1}$ to ~100 m$^3$ s$^{-1}$ in the Trinity River) with SPM generally reduced throughout the bay. However, Trinity Bay and East Bay showed relatively higher SPM than the upper and lower Galveston Bay (Fig. 9g), likely due to a more delayed discharge through the Trinity River and the wetlands.

Overall, the QAA-V based SPM maps of Galveston Bay showed distinct variations in SPM concentrations following Hurricane Harvey. Although the Trinity Bay and the upper Galveston Bay responded similarly a few days following the hurricane, distinct SPM patterns emerged (e.g., lower SPM in the western and

higher in the eastern part of the bay) just after two weeks, suggesting different influences in the eastern and western parts of the bay. For example, flood waters from the Houston metropolitan and surrounding region appeared to have receded within a few days of the hurricane event (Fig. 9a), whereas the discharge of flood waters were elevated through the Trinity River over the course of several weeks after Hurricane Harvey. These flood waters could have accumulated first in wetlands and numerous water bodies in the eastern region and lower Trinity Basin during the hurricane event and eventually emptied to the main Trinity River channel. SPM was also elevated in the East Bay, but concentrations were generally lower within the within the first two weeks suggesting this region of the bay mostly remained isolated from the other regions of Galveston Bay. Nonetheless, this region receives discharge from the surrounding wetlands which could have been elevated during this period. Overall, wind forcing was also important in controlling the extent and the dispersal of the sediment-rich plume waters into the shelf and in contributing to the SPM variability within the bay due to sediment resuspension and its transport into the shelf waters.

## 4.4 Application of the QAA tuning to various ocean color and land observing sensors

Sensor-specific QAA tuning (e.g., QAA-V for VIIRS, MODIS, Landsat-8 OLI, and Sentinel3 OLCI) showed overall valid retrieval of absorption and backscattering coefficients with various ocean color and land-observation sensors (Fig. 10). Although satellite-derived values and trends of $a_{tnw}443$ and $b_{btnw}470$ are similar to the field observations, the observed discrepancies could in addition to the uncertainties in field measurements, be due to several sources of errors. For example, it is well-known that satellite products suffer from large errors in the blue region especially in coastal waters due to atmospheric correction (Table 4; Supplementary S4). The large errors between field and Landsat8 OLI derived $a_{tnw}443$ could have been due to the fact that the QAA processing chain uses these erroneous blue Rrs values to obtain $a_{tnw}$ at the blue wavelengths (Table 2; Level 3). However, the $b_{btnw}$ retrievals at the blue wavelengths are unaffected by the blue Rrs inputs (Table; Level 2). Likewise, the errors were relatively smaller at the reference wavelength (Table 4) because the proposed QAA tuning avoided using blue wavelengths in the primary step of getting $a_{tnw}$ and $b_{btnw}$ at a reference wavelength. Hence, the atmospheric correction procedure is an important step that would impact the performance of QAA-V in ocean color applications of shallow estuarine and near-shore waters.

Another important discrepancy among various sensor maps is the number of masked pixels in shelf waters (Fig. 10). We have used a VIIRS band-ratio based threshold ($\rho=0.65$) to separate green waters (e.g., productive coastal waters) and blue waters (e.g., open ocean) (Eq. 15) and applied it to various satellite sensors.

Although this threshold worked well for the sensors with similar green and red bands (e.g., MODIS-A and VIIRS), it did not perform as well for Sentinel 3 OLCI and Landsat 8 OLI because of notable differences between the green and red bands of these sensors and the VIIRS sensor (Table 3). Thus, while the proposed threshold works well to represent estuarine and near-shore waters for various ocean color and land-observing sensors, it could be further optimized for each satellite sensor.

## 5 Conclusions

A multiband quasi-analytical algorithm tuned for the VIIRS ocean color sensor (QAA-V) and for the estuarine and near-shore waters was proposed. Two major changes were applied to the standard QAA (Lee et al., 2002), 1) the coefficients $g_0$ and $g_1$ of a semi-analytical quadratic relationship were updated to obtain $u$ from the Rrs (Eq. 1), and 2) a threshold-based empirical model was proposed using the green to red band ratio (GRBR) to estimate the total absorption coefficient at a reference wavelength. The QAA-V derived total absorption and backscattering coefficients showed a good relationship in a variety of waters ranging from highly turbid and highly absorbing (MRE<17 %) to relatively clearer coastal waters (MRE<30 %). Moreover, a reasonable performance (MRE<25 %) using in-situ estuarine and near-shore data indicated the usefulness of the GRBR in modeling total absorption coefficients in estuarine waters regardless of the dominance of one or more water constituent (e.g., CDOM, mineral particles, or phytoplankton). This band ratio needs to be explored further for various ocean color sensors and in different estuarine and near-shore environments around the world. The QAA-V may not perform satisfactorily in optically shallow waters as the empirical relationships were designed specifically for the optically deep environments. This study showed good retrieval of backscattering coefficients (MRE=~25–30 %); however, the errors increased towards the lower levels of the QAA-V processing chain (e.g., MRE=~35 % in estimating SPM) likely due to the lack of $bb_{tnw}$–SPM matchups in variety of waters in formulating a robust SPM–$b_{btnw}$532 relationship corresponding to estuarine waters. This limitation suggests a great need of in-situ backscattering measurements in various shallow environments and their availability for the public use.

The QAA-V and a regional SPM–$b_{btnw}$532 relationship were applied to a sequence of VIIRS imagery for investigating post-hurricane SPM dynamics in Galveston Bay. Despite noticeable errors, especially in turbid regions of the bay, the application of QAA-V showed great potential in revealing the SPM patterns due to post-hurricane variations linked to freshwater inflow to the bay and wind forcing. A sequence of SPM maps after the passage of Hurricane Harvey showed that Galveston Bay received massive amounts of SPM due to large volumes

of freshwater inputs from the two major rivers and surrounding regions. However, while the freshwater inflow reduced within a few days in the western part of the bay, it remained high over the course of several weeks in the eastern part of the bay providing evidence of the short-term storage capacity of wetlands and numerous freshwater reservoirs in the lower Trinity Basin. This discharge pattern coupled with different residence times in the western and eastern parts of the bay resulted in distinct SPM patterns in the two regions of the bay. Furthermore, while winds appeared to have played an important role in re-suspending sediments within the bay, they were critical in the transport and dispersion of sediments into the shelf waters of the Gulf of Mexico.

This study did not address bottom reflectance effects (e.g., during clear water conditions), Raman scattering, and chlorophyll fluorescence that may have degraded the QAA-V's performance. Furthermore, the tuning of the semi-analytical and empirical relationships (e.g., η in Table 2) was not possible due to scarcity of field IOPs (e.g., backscattering measurements). Although, further refinements and validation studies are necessary to improve the performance and applicability of QAA-V in spatially and temporally distinct shallow waters around the world, the promising results of this study suggest that the application of QAA-V to various ocean color and land observation satellites can be a useful tool to assess the bio-optical state and water quality dynamics in a variety of coastal systems around the world.

**Data availability**

Data from field measurements are available upon request from the corresponding author.

**Competing interests**

The authors declare that they have no conflict of interest.

**Acknowledgements**

We thank NASA Ocean Color Biology Processing Group (OBGP) for providing access to the VIIRS ocean color data. We also acknowledge NASA OBGP for SeaBASS data repository, and individual project PIs who made their bio-optical observations available for public use. We thank USGS and NOAA for providing various meteorological and hydrological data to support this work. The authors acknowledge NASA funding through grant No. 80NSSC18K0177. We are also thankful to two anonymous reviewers for their insightful comments and suggestions that greatly improved this manuscript.

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

**Table 1: In-situ estuarine & near-shore dataset (IES: IOPs and Rrs matchups), compiled from the SeaBASS, with locations, possessing status, project instructors, and the purpose in this analysis. Apalachicola Bay, Barataria Bay, and Galveston Bay datasets were collected by the authors during various field surveys.**

| Experiment | Location (Depth <10 m) | N | Processing status | Investigator(s) | Purpose for QAA-V |
|---|---|---|---|---|---|
| SWFL | South-west Florida | 5 | Final | Chuanmin Hu | Validation |
| EcoHAB | South-west Florida | 74 | Final | Kendall Carder | Tuning |
| Chesapeake Bay – Light Tower | Chesapeake Bay | 8 | Preliminary | Richard Zimmermann & Glenn Cota | Validation |
| GEO-CAPE | Chesapeake Bay | 19 | Final | Richard Miller | Validation |
| Tampa Bay | Tampa Bay | 47 | Final | Chuanmin Hu | Tuning |
| Lake Erie | Lake Erie | 3 | Preliminary | Rick Gould | Validation |
| Horn Island | Horn Island | 6 | Preliminary | Bob Arnone | Validation |
| CoJet-4,5,6 | Mobile Bay | 18 | Preliminary | Don Johnson | Validation |
| Cojet-7 | Mississippi Sound | 6 | Preliminary | Rick Gould | Validation |
| BluCAR | Apalachicola Bay | 32 | Final | Eurico D'Sa & Christopher Osburn | Validation |
| BluCAR | Barataria Bay | 31 | Final | Eurico D'Sa & Christopher Osburn | Validation |
| Bio-optics | Chesapeake Bay | 43 | Final | Alex Gilerson | Validation |
| SFP | Florida Bay | 8 | Final | Frank Muller-Karger | Validation |
| Hurricane Harvey | Galveston Bay | 27 | Final | Eurico D'Sa & Ishan Joshi | Validation |
| GEOCAPE GOMEX | Northern Gulf of Mexico shelf | 13 | Final | Antonio Mannino & Michael Novak | Validation |

**Table 2: Processing steps of QAA-V for obtaining total absorption ($a_t$) and backscattering coefficients ($b_{b_t}$). Levels 0 and 1C were adopted from Lee et al. (2002), whereas 1A and 1B were modified in this study[1]. Level 1C1 is adopted from D'Sa et al. (2007).**

| Level | Parameter | Model | Type |
|---|---|---|---|
| 0 | $R_{rs}^{0-}(\lambda)$ | $R_{rs}^{0-}(\lambda) = \dfrac{R_{rs}^{0+}(\lambda)}{0.52 + 0.17 \times R_{rs}^{0+}(\lambda)}$ | Semi-analytical |
| 1A | $u(\lambda)$ | $u(\lambda) = \dfrac{-g_0 + [g_0^2 + 4 \times g_1 \times R_{rs}^{0-}(\lambda)]^{0.5}}{2 \times g_1}$ <br> $\rho = \log_{10}\left(\dfrac{R_{rs}^{0-}(\lambda_0)}{R_{rs}^{0-}(671)}\right)$ <br> $g_0 = 0.0788$ and $g_1 = 0.2379$ for $\rho < 0.25$ <br> $g_0 = 0.0895$ and $g_1 = 0.1247$ for $\rho \geq 0.25$ | Semi-analytical |
| 1B | $a_{tnw}(\lambda_0)$ <br> $\lambda_0 = 551$ or $555$ <br> $\lambda_1 = 671$ | $a_{tnw}(\lambda_0) = \begin{cases} 10^{(0.139 - 1.788 \times \rho + 0.490 \times \rho^2)} & \text{if } \rho < 0.25 \\ 10^{(0.406 - 2.940 \times \rho + 0.928 \times \rho^2)} & \text{if } \rho \geq 0.25 \end{cases}$ <br> $\rho = \log_{10}\left(\dfrac{R_{rs}^{0-}(\lambda_0)}{R_{rs}^{0-}(\lambda_1)}\right)$ | Empirical |
| 1C 0 | $b_{b_{tnw}}(\lambda_0)$ | $b_{b_{tnw}}(\lambda_0) = \dfrac{(a_{tnw}(\lambda_0) + a_w(\lambda_0)) \times u(\lambda_0)}{1 - u(\lambda_0)} - b_{b_w}(\lambda_0)$ | Analytical |
| 1 | $\eta$ | $\eta = -0.566 - 1.395 \times \log_{10}(b_{btnw}555)$ | Empirical |
| 2 | $b_{b_t}(\lambda)$ | $b_{b_t}(\lambda) = b_{b_w}(\lambda) + b_{b_{tnw}}(\lambda_0) \times \left(\dfrac{\lambda_0}{\lambda}\right)^{\eta}$ | Semi-analytical |
| 3 | $a_t(\lambda)$ | $a_t(\lambda) = b_{b_t}(\lambda) \times \left(\dfrac{1 - u(\lambda)}{u(\lambda)}\right)$ | Analytical |
| SPM models | | | |
| This study | | $SPM = (103.07 \times b_{b_{tnw}}532) + 0.24$ | Empirical |
| D'Sa et al. (2007) | | $SPM = (106.93 \times b_{b_{tnw}}555) + 0.61$ | Empirical |
| Nechad et al. (2010) | | $SPM = \left(\dfrac{A^{\rho} \times \rho_w}{1 - \rho_w/C^{\rho}}\right) + B^{\rho}$; <br> where $A^{\rho}=373.79$ mg L$^{-1}$, $B^{\rho}=1.47$ mg L$^{-1}$, $C^{\rho}= 0.1747$ for $\lambda$=670 nm | Empirical |

[1]Parameters $g_0 = 0.0788$ and $g_1 = 0.2379$ were derived with HL datasets. $\lambda_0= 551$ or $555$nm; $R_{rs}^{0-}$ = remote-sensing reflectance just below water surface; $a_{tnw}$ = total non-water absorption coefficient; $b_{b_{tnw}}$= total non-water backscattering coefficient; $a_w$ = water absorption coefficient; $b_{b_w}$= water backscattering coefficient; $\eta$ = power-law exponent (D'Sa et al., 2007), SPM = suspended particulate matter concentration. Note: In SPM model comparison, input backscattering values are obtained from QAA-V, whereas surface reflectance ($\rho$) is obtained by multiplying $\pi$ and above-surface remote sensing reflectance ($Rrs^{0+}$)

**Table 3: The calibration coefficients for sensor-specific QAA tuning. $\lambda_0$ is a sensor-specific reference wavelength.**

| Sensor | $\rho = \log_{10}\left(\dfrac{R_{rs}^{0-}(\lambda_0)}{R_{rs}^{0-}(\lambda_1)}\right)$ | $a_{tnw}(\lambda_0) = 10^{(a+b\times\rho+c\times\rho^2)}$ (Level 1B —Table 2) | | | | | |
|---|---|---|---|---|---|---|---|
| | | $\rho < 0.25$ | | | $\rho \geq 0.25$ and $\rho \leq 0.65$ | | |
| | | a | b | c | a | b | c |
| VIIRS | $\lambda_0 = 551$ nm & $\lambda_1 = 671$ nm | 0.139 | -1.788 | 0.490 | 0.406 | -2.940 | 0.928 |
| MODIS-Aqua | $\lambda_0 = 555$ nm & $\lambda_1 = 667$ nm | 0.091 | -1.800 | 0.560 | 0.275 | -2.674 | 0.813 |
| Sentinel3 OLCI | $\lambda_0 = 560$ nm & $\lambda_1 = 674$ nm | 0.176 | -1.830 | 0.528 | 0.397 | 2.940 | 0.800 |
| MERIS | $\lambda_0 = 560$ nm & $\lambda_1 = 665$ nm | 0.081 | -1.868 | 0.688 | 0.314 | -2.733 | 0.713 |
| SeaWiFS | $\lambda_0 = 555$ nm & $\lambda_1 = 670$ nm | 0.128 | -1.792 | 0.505 | 0.276 | -2.742 | 0.842 |
| Sentinel2 MSI | $\lambda_0 = 560$ nm (Band 3) & $\lambda_1 = 665$ nm (Band 4) | 0.0814 | -1.868 | 0.688 | 0.223 | -2.732 | 0.740 |
| Landsat8 OLI | $\lambda_0 = 560$ nm (Band 3) & $\lambda_1 = 655$ nm (Band 4) | -0.087 | -1.900 | 0.952 | 0.057 | -2.667 | 0.753 |

**Table 4: The performance evaluation of the atmospheric correction procedure (MUMM) in Galveston Bay (USA) during two field campaigns (September 29, 2017 & October 29–30, 2017).**

| Image date | Difference of days for field observation | Number of stations | Absolute Mean relative error (MRE) (%) | | | | | |
|---|---|---|---|---|---|---|---|---|
| | | | 410 nm | 443 nm | 486 Nm | 551 nm | 671 nm | All bands |
| September 30, 2017 | +1 | 9 | 114.7 | 40.5 | 19.8 | **10.9** | **18.1** | 40.8 |
| October 29, 2017 | 0 | 10 | 55.8 | 12.4 | 5.1 | **6.3** | **8.9** | 17.7 |
| October 30, 2017 | 0 | 7 | 139.6 | 42.1 | 21.5 | **7.5** | **7.6** | 43.7 |

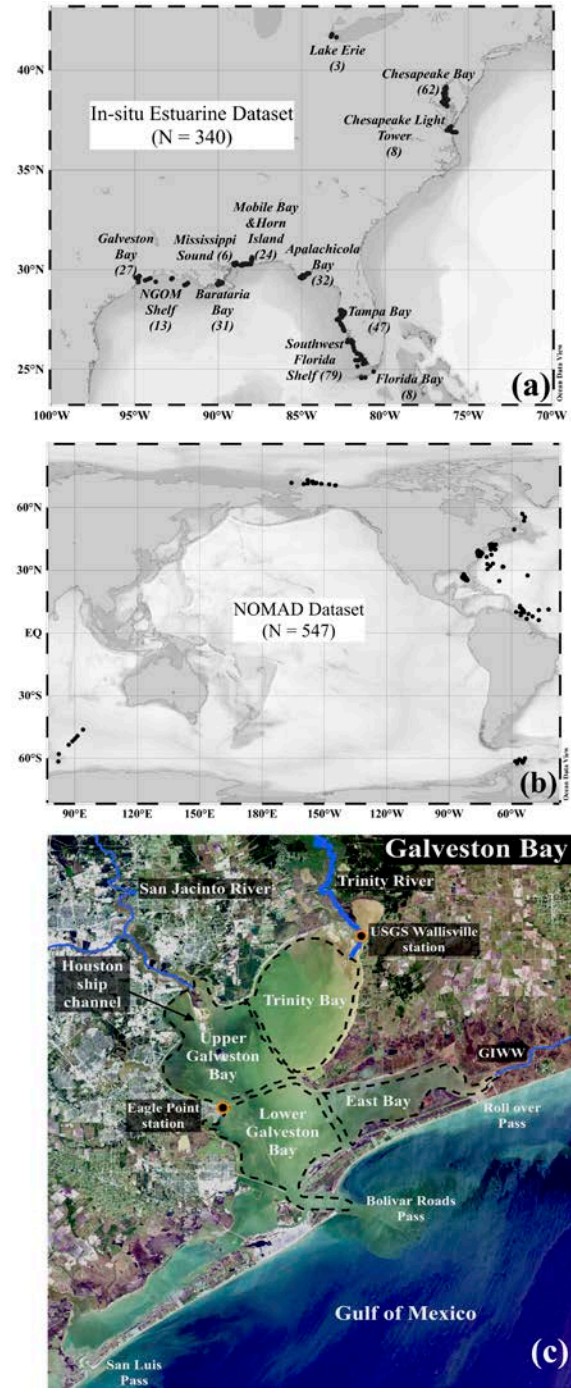

**Figure 1: a) In-situ dataset (IES) representing estuarine and near-shore waters (< 10 m) in the U.S. East Coast and northern Gulf of Mexico (N=340), b) NOMAD dataset (N=547), and c) Galveston Bay, Texas (USA). GIWW=Gulf Inter-Coastal Water Way.**

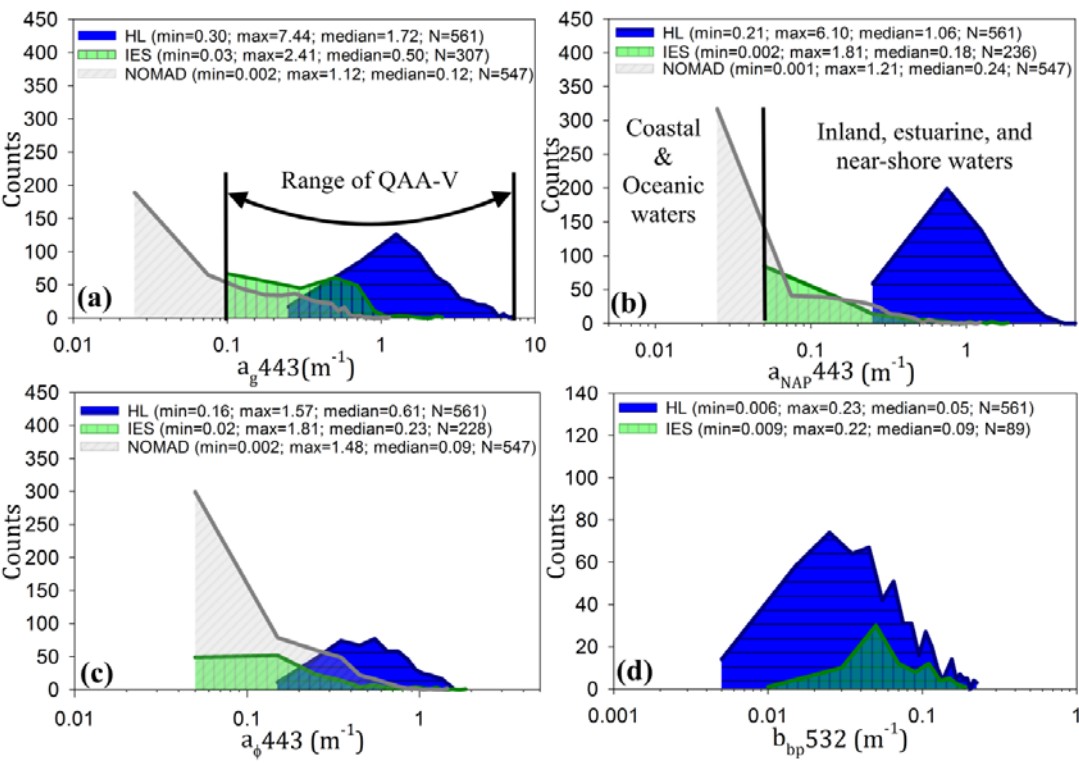

**Figure 2: Data statistics and distribution of water IOPs, a) $a_g443$, b) $a_{NAP}443$, c) $a_\phi443$, and d) $b_{btnw}532$, for synthetic data (HL - Hydrolight®; blue color), in-situ estuarine & near-shore data (IES; green color), and NOMAD data set (NOMAD; grey color). Range of QAA-V indicates data that are used to update QAA-V for shallow waters (*e.g.*, HL and IES).**

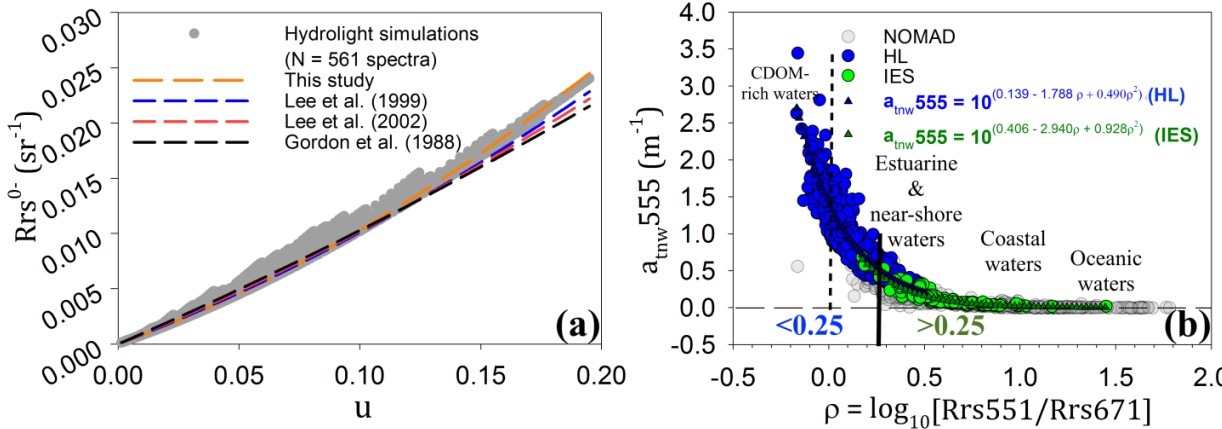

**Figure 3.** a) u (= $b_b/(a+b_b)$) vs. $R_{rs}^{0-}$ for various studies (this study–orange, Lee et al. (1999)–blue, Lee et al. (2002)–red, and Gordon et al. (1988)–black) using HL datasets, b) a relationship between green to red band ratio and $a_{tnw}555$ for different dataset (HL–blue, IES–green, and NOMAD–grey). Black line shows a threshold to facilitate a smooth transition from in-situ to synthetic data in modeling of $a_{tnw}555$. A dashed line is separating data with negative $\rho$.

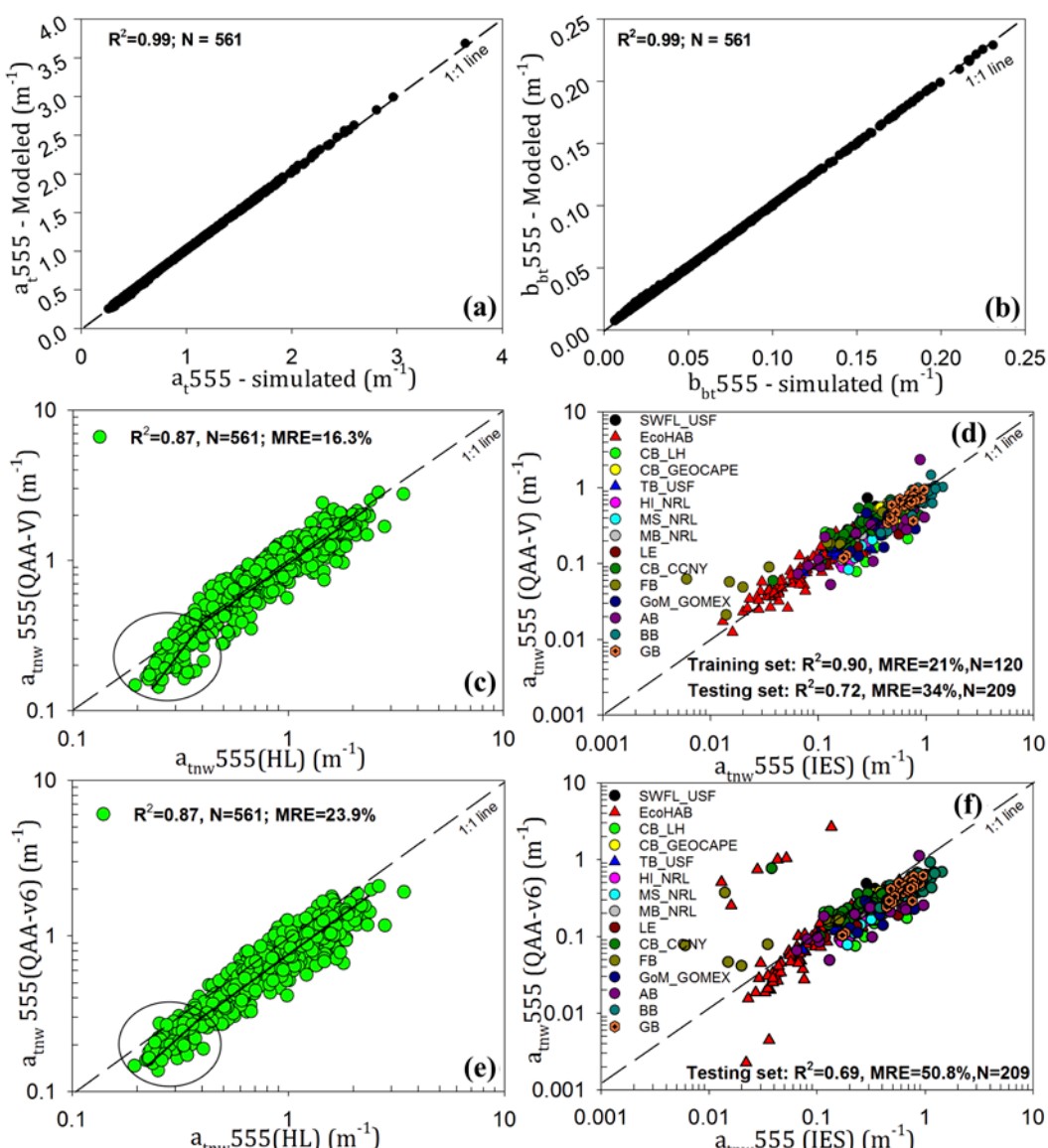

**Figure 4. Validation of Rrs$^{0-}$ vs. u model using HL data for, a) a$_t$555 and b) b$_{bt}$555. Validation of QAA-V modeled a$_{tnw}$555 with "true" a$_{tnw}$555 for c) HL synthetic dataset (bias$_{log10}$=−0.0208, RMSE$_{log10}$=0.0963 m$^{-1}$) and d) IES dataset (bias$_{log10}$=−0.0294, RMSE$_{log10}$=0.190 m$^{-1}$). Validation of QAA-v6 modeled a$_{tnw}$555 with "true" a$_{tnw}$555 for e) HL synthetic dataset (bias$_{log10}$=−0.1180, RMSE$_{log10}$=0.1490 m$^{-1}$) and f) IES dataset (bias$_{log10}$=−0.1252, RMSE$_{log10}$=0.278 m$^{-1}$). [Note: Training set includes EcoHAB and Tampa Bay data (Table 1). IOP measurements of Galveston Bay (this study) are shown with orange hexagons. MRE= absolute mean relative error (%). TB=Tampa Bay, AB=Apalachicola Bay, BB=Barataria Bay, CB=Chesapeake Bay, FB=Florida Bay, GoM=Gulf of Mexico, HI=Horn Island, LE=Lake Erie, MS=Mississippi Sound, MB=Mobile Bay, SWFL=Southwest Florida, GB=Galveston Bay (Table 1).]**

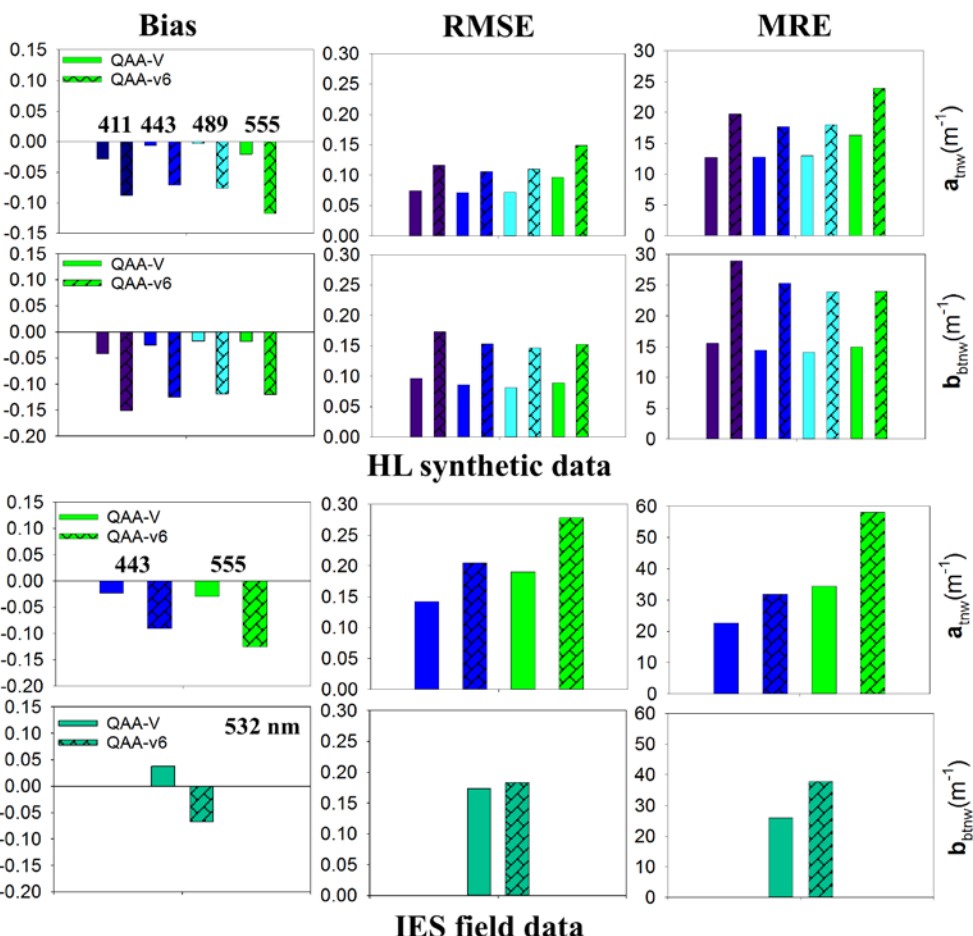

 **Figure 5: Statistical assessment of QAA-V in comparison to QAA-v6 (updated on May 2015) using synthetic data (HL) (top two panel) and estuarine & near-shore field data (IES) (bottom two panels). Color scheme indicates $a_{tnw}$ and $b_{btnw}$ at 411 nm (dark blue), 443 nm (blue), 489 nm (cyan), 555 nm (green), and 532nm (dark green) wavelengths. MRE=absolute mean relative error (%). RMSE and bias are in m$^{-1}$.**

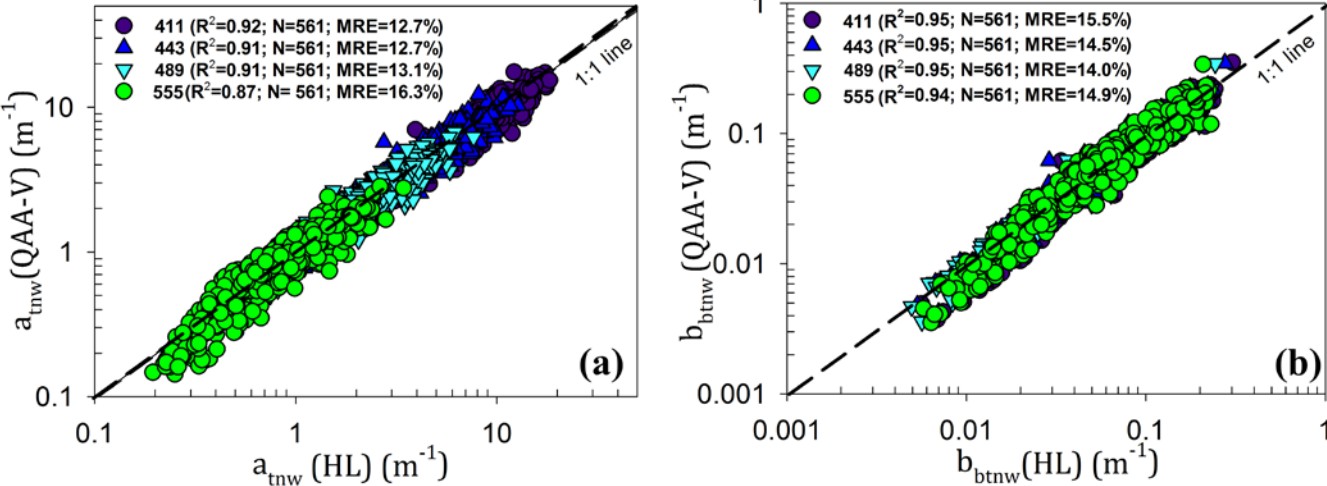

**Figure 6: Validation of QAA-V on synthetic data (HL) for a) $a_{tnw}$ and b) $b_{btnw}$ at 411 nm (dark blue), 443 nm (blue), 489 nm (cyan), and 555 nm (green) wavelengths. MRE=absolute mean relative error (%).**

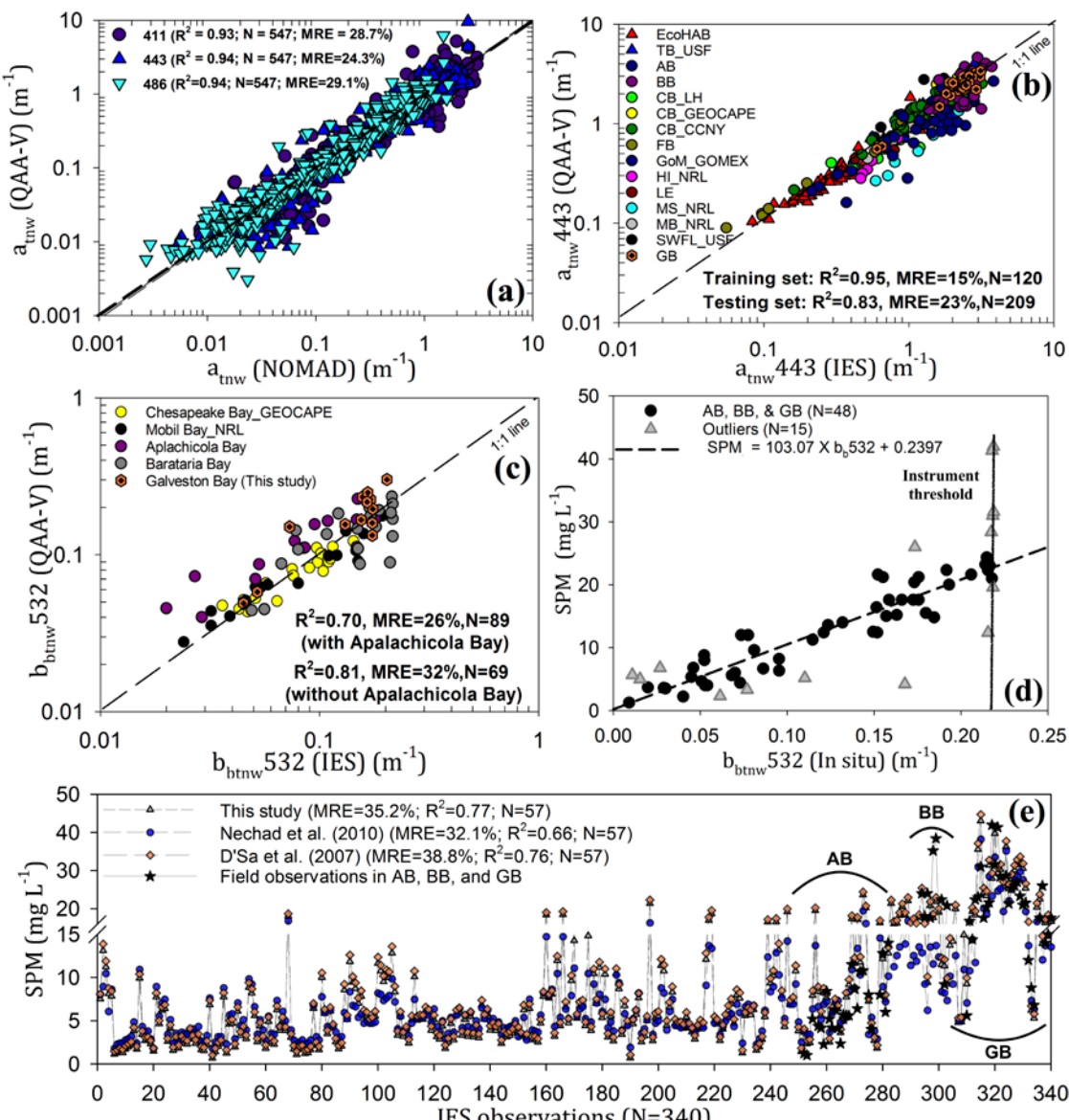

**Figure 7: Evaluation of QAA-V's performance on estuarine & near-shore dataset (IES): a) $a_{tnw}443$ and b) $b_{btnw}532$, and on NOMAD datasets: c) $a_{tnw}$ at 411 nm (dark blue), 443 nm (blue), and 488 nm (cyan) wavelengths. d) SPM$-b_{btnw}532$ relationship ($R^2$ = 0.89 without outliers) that was formulated based on field observations in Apalachicola Bay, Barataria Bay, and Galveston Bay.**
5  **e) A comparison of SPM$-b_{btnw}532$ relationship of this study (grey triangles) with SPM models from Nechad et al. (2010) (blue circles) and D'Sa et al. (2007) (orange squares), and with field observations in AB, BB and GB (black stars). TB=Tampa Bay, AB=Apalachicola Bay, BB=Barataria Bay, CB=Chesapeake Bay, FB=Florida Bay, GoM=Gulf of Mexico, HI=Horn Island, LE = Lake Erie, MS=Mississippi Sound, MB=Mobile Bay, SWFL=Southwest Florida, GB=Galveston Bay (Table 1). Training set includes EcoHAB and Tampa Bay datasets. MRE=absolute mean relative error (%).**

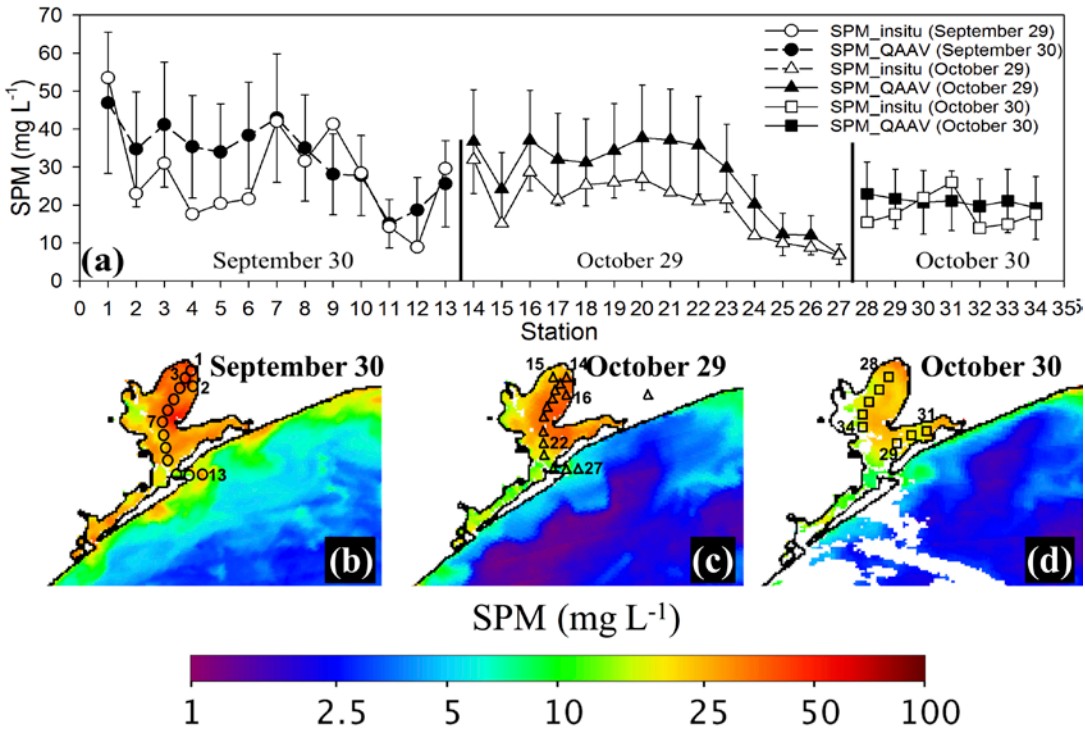

**Figure 8: a) Validation of QAA-V derived SPM vs. in-situ SPM for b) September 30, 2017 (MRE=39.9 %, N=13), c) October 29, 2017 (MRE=39.1%, N=14), and d) October 30, 2017 (MRE=26.6 %, N=7). Stations for validation analysis (Fig. 8a) are also illustrated in corresponding SPM maps.**

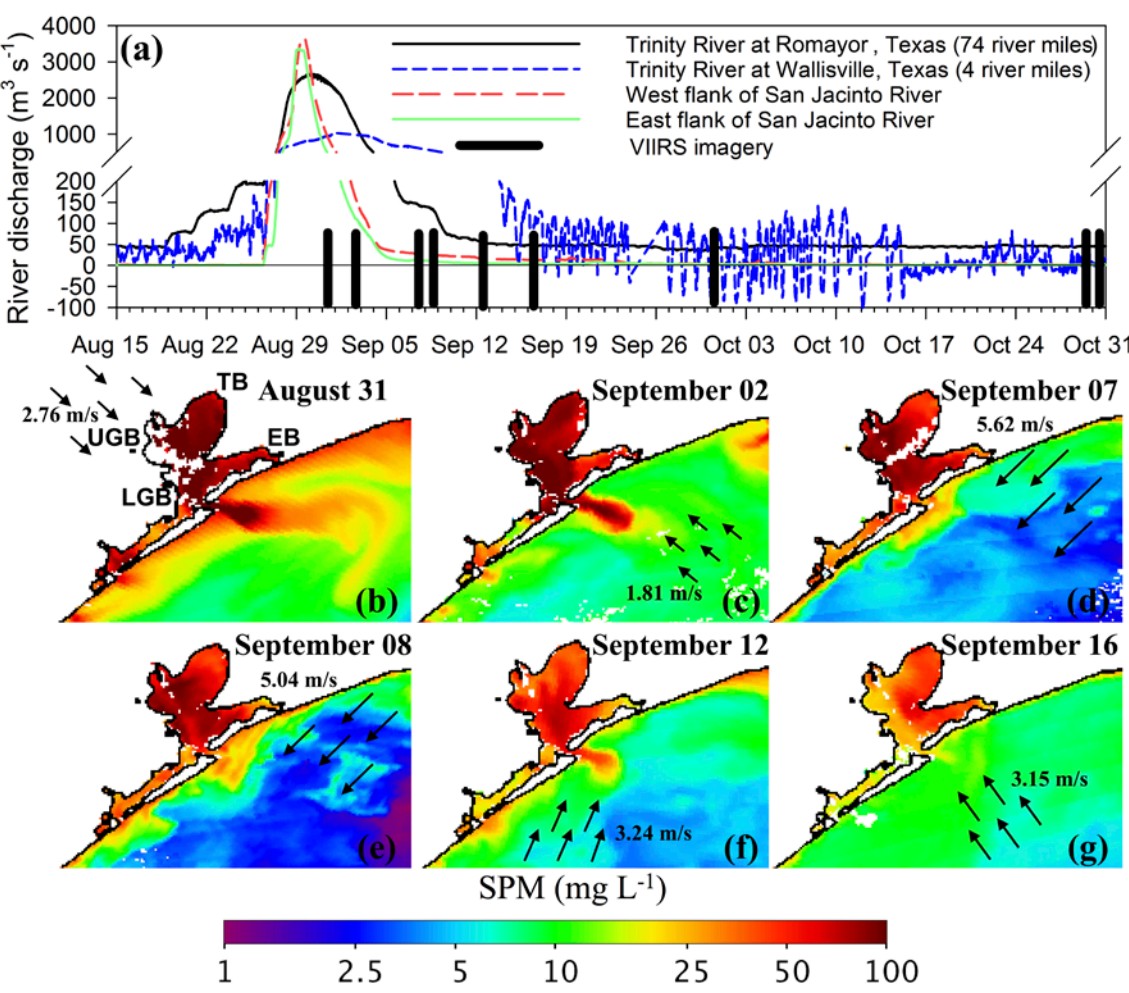

**Figure 9: a) River discharge of the Trinity River at Romayor, Texas (black line) and at Walllisville, Texas (blue line), and the west flank of the San Jacinto River (red line) and the east flank of the San Jacinto River (green line). The river discharge at Wallisville site was not corrected for the tides. Black bars represent the days of VIIRS satellite imagery for SPM analysis in Galveston Bay. b-**
10 **g) Post-hurricane SPM maps of Galveston Bay. TB=Trinity Bay, UGB=Upper Galveston Bay, LGB=Lower Galveston Bay, and EB=East Bay (Fig. 1c).**

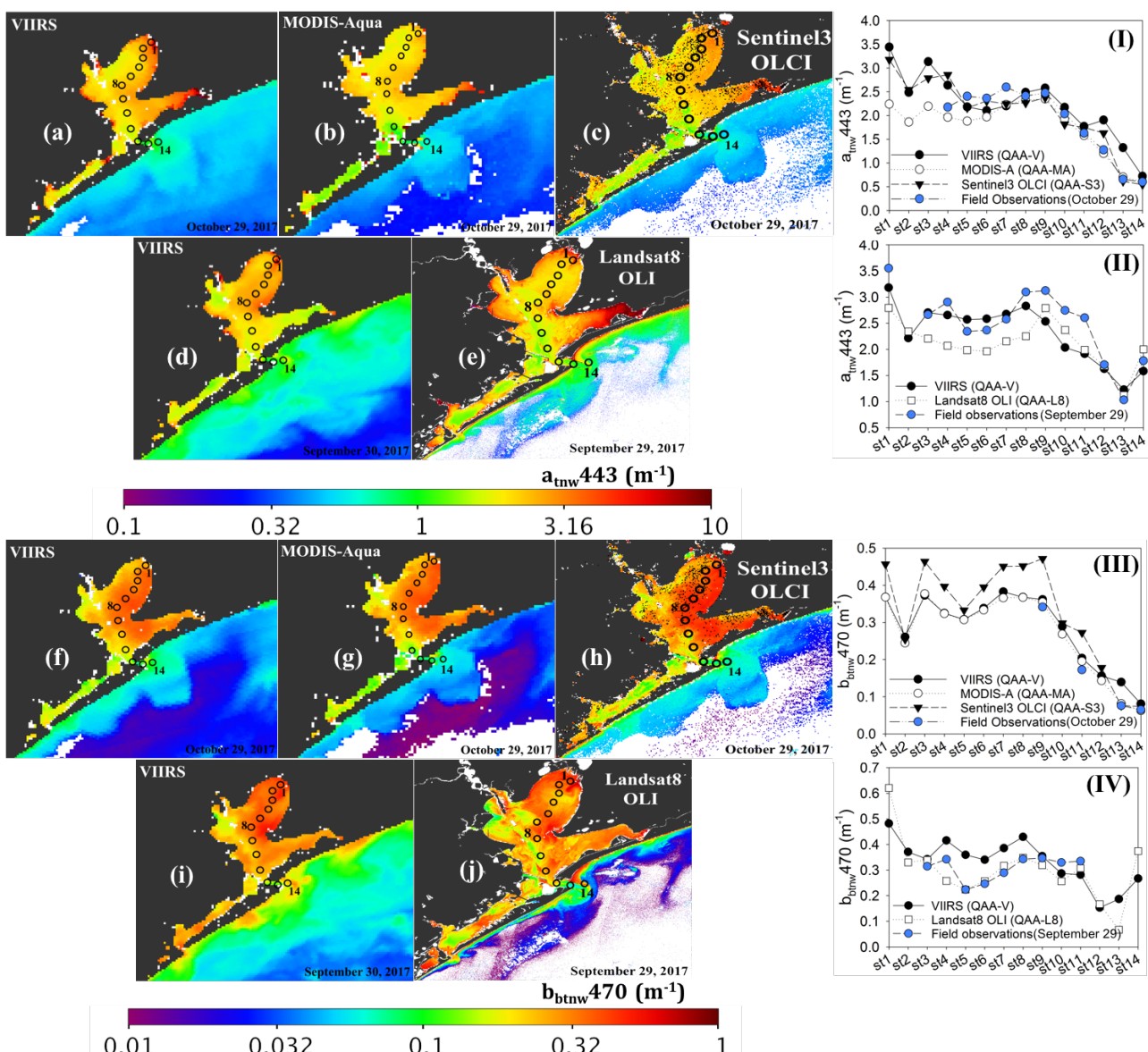

**Figure 10: Application of sensor-specific QAA-V tuning to obtain the maps of $a_{tnw}443$ using a) VIIRS, b) MODIS-Aqua and c) Sentinel3 OLCI on October 29, 2017, and d) VIIRS and e) Landsat8 OLI on September 30, 2017 and September 29, 2017, respectively. The validation of these maps with the field observations along the transect (St. 1 to St. 14) is shown in (I) for figs. 8a-8c and in (II) for figs. 8d & 8e. The maps of $b_{btnw}470$ were obtained similarly for f) VIIRS (October 29, 2017), g) MODIS-Aqua, h) Sentinel3 MSI, i) VIIRS (September 30, 2017), and j) Landsat8 OLI (September 29, 2017) with their validation results in (III) and (IV), respectively. Parameter values beyond the upper limit of tuned QAA ($\rho > 0.65$) are shown masked in white.**