# Peer review of "An estuarine tuned Quasi-Analytical Algorithm for VIIRS (QAA-V): assessment and application to satellite estimates of SPM in Galveston Bay following Hurricane Harvey"

_Biogeosciences, 2018_

## Referee Comment (RC1) · Anonymous Referee #1 · 21 May 2018

The main objective of the paper is to propose an improved version of the Quasi-Analytical Algorithm for the VIIRS ocean color sensor (QAA-V) and for estuarine and near-shore water applications. Calibration and validation of the QAA-V are based on a large synthetic and in situ dataset. Results are convincing. I particularly appreciate the effort intended to present and motivate the modifications/improvements of the standard QAA. Otherwise, I think the paper is well written, clear and very readable. However, I note three major deficiencies before publication. By consequence, I recommend this manuscript for publication in Biogeosciences but only after minor revisions are made in

order to address the following comments : 1- The QAA-V was developed for the VIIRS ocean color sensor. I find that the VIIRS-specific development of the QAA-V limits the scope of the study. Moreover, authors do not motivate the choice of this sensor. For example, why choose VIIRS while Landsat-8/OLI or Sentinel-2/MSI provide data with a better spatial resolution (which is crucial for coastal applications)? I recommend to the authors to make explicit the choice of VIIRS. I also recommend that authors provide in a table the calibration coefficients for other ocean color sensors. 2- Authors mention that the QAA-V can be applied in optically shallow waters. For instance, p.1, lines 8-11 : "The standard quasi-analytical algorithm (Lee et al., 2002) was tuned as QAA-V using a suite of synthetic data and in-situ measurements to improve its performance in OPTICALLY complex and shallow estuarine waters". p.4, lines 5-7 : "In this study, we present a tuned multiband Quasi-Analytical Algorithm (QAA-V) optimized to estimate IOPs in OPTICALLY shallow and near-shore waters for the Visible and Infrared Imaging Radiometric Suite (VIIRS) ocean color sensor". or, p.19, lines 12-14 : "The QAA-V may not perform satisfactorily in optically deep waters as the empirical relationships were designed specifically for the optically shallow environments". I think this error comes from a lack of knowledge of the authors of the definition of "optically shallow waters". "Optically shallow waters" doesn't mean "shallow waters". A definition can be found in the IOCCG Report Number 3 (2000). "Optically shallow implies that the product of the diffuse attenuation coefficient Kd and the geometric depth z is small" (p.33). "Coastal waters can also be optically shallow, so that water-leaving radiance is affected by bottom reflectance" (p.94). "Where coastal waters are optically shallow, algorithms for water-column constituents need to remove contributions from bottom reflectance" (p.99). For highly absorbing and turbid waters (which is the case of this study), we can expect a high value of Kd and consequently a high value of the product of Kd and z (even in the case where z is small). It is difficult to believe that the water-leaving radiance is significantly affected by bottom reflectance. More important, the QAA is not designed to take into account the contribution from bottom reflectance. No study has ever shown that QAA works in optically shallow waters. By consequence,

I recommend to the authors to replace "optically shallow waters" by "shallow waters". Moreover, for clarity, the author should also specify that the QAA-V was developed for optically deep waters. 3- The results do not really demonstrate the interest of using QAA rather than existing algorithms (for instance, Nechad et al. (2010) or Han et al. (2016)) to estimate SPM. P.16, lines 15-22, authors mention the limits of the use of Rrs to estimate SPM before to underline the interest of the use of bbp. They forget to mention the strong limits of this alternative method. bbp is not directly measured. The inversion model used to derivate bbp generates an inherent error that propagates for the SPM inversion. Another source of error is due to the fact that the bbp to SPM ratio is not constant and its value depends of the particle nature. I recommend to authors to discuss precisely the limits of the "bbp method" for the SPM estimation.

---

## Referee Comment (RC2) · Anonymous Referee #2 · 31 May 2018

General comment

This study presents a tuning of the well-known standard quasi-analytical algorithm (QAA, Lee et al., 2002) to improve its performance in optically complex and shallow estuarine waters. This tuning is based on both synthetic data and in-situ measurements. The improvement due the tuned algorithm (called QAA-V) is assessed based on once again synthetic data (Hydrolight computations) and in-situ measurements in several US coastal and estuarine waters. The performance of the tuned algorithm is proved to be reasonable so as the improvement when compared to QAA last version

(v6). Finally, QAA-V is applied to VIIRS satellite data recorded over turbid coastal waters (Galveston Bay, Texas) to retrieve then map the particulate light backscattering coefficient at 532 (bbp532) actually converted into suspended particulate matter concentrations (SPM) using an empirical relationship. SPM maps generated after an intense flooding event are analyzed using river discharge and wind speed/direction data. Overall, the manuscript presents a significant piece of work with interesting results. Moreover, it is well organized and written. The most convincing part is certainly the tuning of the QAA algorithm for shallow estuarine waters and the efforts made to validate the results obtained. The less convincing section is the retrieval and mapping of SPM concentrations in Galveston Bay using VIIRS satellite data as: QAA allows retrieving the light absorption and particulate coefficients of colored water constituents on top of pure water (anw and bbnw, respectively), i.e. much more than SPM concentration. One would expect maps of anw and bbnw to be presented and analyzed at the end of the study that could highlight interesting spatial and temporal variations of CDOM and SPM composition and size distribution over the study area.

My recommendation is to accept the manuscript for publication if the authors can address in more details this last comment, at least in the Discussion and Conclusions sections.

Additional comments

Page 6 SPM uncertainty? Sensor used for bbp measurements and data processing/corrections?

Eq. 2: Rrs(residual)? Please define and justify

Eqs. 1-3: provide physical unit for each parameter
* * *

---

## Author Comment (AC1) · 7 Jun 2018

The authors would like to thank Reviewer #1 for the detailed and helpful comments that will improve the quality and clarity of the paper. The author's responses are detailed below in italics.

**Response to Anonymous Reviewer #1**

"The main objective of the paper is to propose an improved version of the Quasi-Analytical Algorithm for the VIIRS ocean color sensor (QAA-V) and for estuarine and near-shore water applications. Calibration and validation of the QAA-V are based on a large synthetic and in situ dataset. Results are convincing. I particularly appreciate the effort intended to present and motivate the modifications/improvements of the standard QAA. Otherwise, I think the paper is well written, clear and very readable."

**Response: Thanks. We are glad that the reviewer recognizes our effort to improve the standard QAA.**

"However, I note three major deficiencies before publication. By consequence, I recommend this manuscript for publication in Biogeosciences but only after minor revisions are made in order to address the following comments: 1- The QAA-V was developed for the VIIRS ocean color sensor. I find that the VIIRS-specific development of the QAA-V limits the scope of the study."

Response: We have added calibration coefficients for other ocean color (e.g., MERIS, MODIS-A, Sentinel3 OLCI, and SeaWiFS) and land observing (e.g., Landsat8 OLI and Sentinel2 MSI) sensors in the revised manuscript (Table S1). Additionally, we also showed the performance of the proposed calibration coefficients through performance comparisons between the sensors and field observations (Figure S1). We added Figure 10, Table 3, new sub-sections in the Results and Discussion sections. Also, based on the extension of the QAA-V to a larger number of ocean color sensors, we will modify the title to "An estuarine tuned Quasi-Analytical Algorithm (QAA-V): assessment and application to satellite estimates of SPM in Galveston Bay following Hurricane Harvey"

**Results Section**

**3.6 Extending the QAA-V tuning to additional satellite sensors**

The estuarine-specific green to red band tuning was further applied to evaluate and to extend its applicability to past and present ocean color (e.g., Sentinel3 OLCI, MODIS-Aqua, MERIS, and SeaWiFS) and land observing sensors (Landsat8 OLI and Sentinel2 MSI) (Table 3). The validation analysis showed promising performance of QAA tuning in obtaining total non-water absorption coefficient ( $a_{tnw}443$ ) and total-non water backscattering coefficient ( $b_{btnw}470$ ) in optically complex and shallow waters of Galveston Bay (Fig. 10). Overall, different satellite sensors showed similar trends of  $a_{tnw}443$  and  $b_{btnw}470$  along the transect despite having different spectral and spatial sensor resolutions (Fig. 10a-j;10I-IV). The MRE were ~15 %, 9 %, and 12 % for  $a_{tnw}443$  retrievals from VIIRS, MODIS-A, and Sentinel3 OLCI sensors respectively (Fig. 10a-c & 10I), whereas they were ~26 %, 7 %, 22 % for  $b_{btnw}470$  retrievals on October 29, 2017 (Fig. 10f-h & 10III). For Landsat8 OLI, these MRE were ~20 % and ~10 % for  $a_{tnw}443$  and  $b_{btnw}470$ , respectively on September 29, 2017 (Fig. 10e, 10j, 10II, & 10IV).

**Discussion Section**

**4.4 Application of the QAA tuning to additional ocean color and land observing sensors**

Sensor-specific QAA tuning showed overall valid retrieval of absorption and backscattering coefficients with various ocean color and land observation sensors (Fig. 10). Although satellitederived values and trends of  $a_{tnw}443$  and  $b_{btnw}470$  are similar to the field observations, the observed discrepancies could be attributed to several sources of errors. For example, it is wellknown that satellite products suffers from large errors in the blue region especially due to the atmospheric correction (Supplementary S3). The large errors in IOPs at the blue wavelengths could have resulted due to the fact that the QAA processing chain uses these erroneous Rrs values to obtain  $a_{tnw}$  and  $b_{btnw}$  at the blue wavelengths. Likewise, the errors were relatively smaller at the reference wavelength because the proposed QAA tuning avoided using blue wavelengths in the primary step of getting  $a_{tnw}$  and  $b_{btnw}$  at a reference wavelength (Level 1B in Table 2). Hence, the success of the atmospheric correction procedure is a vital component for using QAA in ocean color application in shallow estuarine and near-shore waters. Further, the uncertainties in field measurements can additionally contribute to this difference.

"Moreover, authors do not motivate the choice of this sensor. For example, why choose VIIRS while Landsat-8/OLI or Sentinel-2/MSI provide data with a better spatial resolution (which is crucial for coastal applications)? I recommend to the authors to make explicit the choice of VIIRS. I also recommend that authors provide in a table the calibration coefficients for other ocean color sensors."

Response: We focused on VIIRS primarily due to our earlier work based on this sensor (Joshi et al. 2017). We appreciate the reviewer's suggestion for using land observing sensors for coastal applications. We worked on several ocean color and land observing sensors and will provide their calibration coefficients and validation analysis in the revised manuscript (Table S1 and Fig. S1). We now strongly feel that our study is applicable to a large number of ocean color sensors, including VIIRS. We will include the following lines in section "2.3 QAA-V processing chain in Materials and methods" to initiate discussion on this additional work,

To extend and to evaluate the applicability of estuarine-specific QAA tuning, it was further applied to various ocean color (Sentinel3 OLCI, MODIS-Aqua, MERIS, and SeaWiFS) and land observation sensors (Landsat8 OLI and Sentinel2 MSI). The calibration coefficients for obtaining total non-water absorption coefficient at a reference wavelength ( $a_{tnw}$  ( $\lambda_0$ ); Level 1B in Table 2) are given in Table 3.

"2- Authors mention that the QAA-V can be applied in optically shallow waters. For instance, p.1, lines 8-11 : "The standard quasi-analytical algorithm (Lee et al., 2002) was tuned as QAA-V using a suite of synthetic data and in-situ measurements to improve its performance in OPTICALLY complex and shallow estuarine waters". p.4, lines 5-7 : "In this study, we present a tuned multiband Quasi-Analytical Algorithm (QAA-V) optimized to estimate IOPs in OPTICALLY shallow and near-shore waters for the Visible and Infrared Imaging Radiometric Suite (VIIRS) ocean color sensor". or, p.19, lines 12-14: "The QAA-V may not perform satisfactorily in optically deep waters as the empirical relationships were designed specifically for the optically shallow environments". I think this error comes from a lack of knowledge of the authors of the definition of "optically shallow waters". "Optically shallow waters" doesn't mean "shallow waters". A definition can be found in the IOCCG Report Number 3 (2000). "Optically shallow implies that the product of the diffuse attenuation coefficient Kd and the geometric depth z is small" (p.33). "Coastal waters can also be optically shallow, so that water-leaving radiance is affected by bottom reflectance" (p.94). "Where coastal waters are optically shallow, algorithms for water-column constituents need to remove contributions from bottom reflectance" (p.99). For highly absorbing and turbid waters (which is the case of this study), we can expect a high value of Kd and consequently a high value of the product of Kd and z (even in the case where z is small). It is difficult to believe that the water-leaving radiance is significantly affected by bottom reflectance. More important, the QAA is not designed to take into account the contribution from bottom reflectance. No study has ever shown that QAA works in optically shallow waters. By

consequence, I recommend to the authors to replace "optically shallow waters" by "shallow waters. Moreover, for clarity, the author should also specify that the QAA-V was developed for optically deep waters."

Response: Thank you for pointing this out. We appreciate the reviewer's detailed comment and suggestion for this error. We will replace "optically shallow waters" by "optically complex and shallow waters" or "shallow waters" at several locations in the revised manuscript.

"3- The results do not really demonstrate the interest of using QAA rather than existing algorithms (for instance, Nechad et al. (2010) or Han et al.(2016)) to estimate SPM. P.16, lines 15-22, authors mention the limits of the use of Rrs to estimate SPM before to underline the interest of the use of bbp. They forget to mention the strong limits of this alternative method. bbp is not directly measured. The inversion model used to derivate bbp generates an inherent error that propagates for the SPM inversion. Another source of error is due to the fact that the bbp to SPM ratio is not constant and its value depends of the particle nature. I recommend to authors to discuss precisely the limits of the "bbp method" for the SPM estimation."

Response: Thank you very much for this suggestion. Because we obtained reasonable estimates of backscattering coefficients from QAA-V processing chain, we decided to include in the latter section of the manuscript (a case study of post-hurricane SPM dynamics in Galveston Bay) as an application of QAA-V for obtaining Level-2 products such as SPM. We strongly agree that there are limitations in the bbp-based approach because of the uncertainty in satellite estimates of bbp (and thus SPM estimates) due to several factors. We listed these limitations in Section 4.2 according to the reviewer's suggestion.

Several factors limit the efficacy of "bbp method" and cause large differences between field and satellite SPM matchups, These include: 1) propagation from various steps of the QAA-V processing chain to  $b_{btnw}532$  (e.g., 20–30 %, Fig. 7c) and hence, further down to the SPM inversion, 2) the uncertainty in the atmospheric-corrected green and red Rrs (e.g., 5–20 %, Table 4), 3) the uncertainty in SPM– $b_{btnw}532$  relationship due to limited observations, 4) the assumption of linearity in SPM– $b_{btnw}532$  model beyond the instrument threshold may not hold well because  $b_{btnw}532$  to SPM ratio depends on the particle characteristics; this may not always be constant especially in highly turbid waters, and 5) errors in SPM measurements.

| wavelength.    |                                                                                         |                                                                              |        |       |       |        |       |  |
|----------------|-----------------------------------------------------------------------------------------|------------------------------------------------------------------------------|--------|-------|-------|--------|-------|--|
|                |                                                                                         |                                                                              |        |       |       |        |       |  |
| Sensor         | $\rho = \log_{10} \left( \frac{R_{rs}^{0-}(\lambda_0)}{R_{rs}^{0-}(\lambda_1)} \right)$ | $a_{tnw}(\lambda_0) = 10^{(a+b\times\rho+c\times\rho^2)}$ (Level 1B—Table 2) |        |       |       |        |       |  |
|                |                                                                                         | $\rho

---

## Author Comment (AC2) · 7 Jun 2018

The authors would like to thank Reviewer #2 for the valuable and detailed comments. The author's responses are in italics.

**Response to Anonymous Reviewer #2**

"This study presents a tuning of the well-known standard quasi-analytical algorithm (QAA, Lee et al., 2002) to improve its performance in optically complex and shallow estuarine waters. This tuning is based on both synthetic data and in-situ measurements. The improvement due the tuned algorithm (called QAA-V) is assessed based on once again synthetic data (Hydrolight computations) and in-situ measurements in several US coastal and estuarine waters. The performance of the tuned algorithm is proved to be reasonable so as the improvement when compared to QAA last version (v6). Finally, QAA-V is applied to VIIRS satellite data recorded over turbid coastal waters (Galveston Bay, Texas) to retrieve then map the particulate light backscattering coefficient at 532 (bbp532) actually converted into suspended particulate matter concentrations (SPM) using an empirical relationship. SPM maps generated after an intense flooding event are analyzed using river discharge and wind speed/direction data. Overall, the manuscript presents a significant piece of work with interesting results. Moreover, it is well organized and written. The most convincing part is certainly the tuning of the QAA algorithm for shallow estuarine waters and the efforts made to validate the results obtained."

Response: *We appreciate that the reviewer found this work interesting and significant.*

"The less convincing section is the retrieval and mapping of SPM concentrations in Galveston Bay using VIIRS satellite data as: QAA allows retrieving the light absorption and particulate coefficients of colored water constituents on top of pure water (anw and bbnw, respectively), i.e. much more than SPM concentration. "One would expect maps of anw and bbnw to be presented and analyzed at the end of the study that could highlight interesting spatial and temporal variations of CDOM and SPM composition and size distribution over the study area. My recommendation is to accept the manuscript for publication if the authors can address in more details this last comment, at least in the Discussion and Conclusions sections."

Response: *Because we obtained reasonable estimates of backscattering coefficients from QAA-V processing chain, we decided to include in latter part of this study an application of QAA-V for investigating post-hurricane SPM dynamics in Galveston Bay. In the revised version, we show results of $a_{tnw}$ and $b_{btnw}$ maps for VIIRS and for additional ocean color and land observing sensors during our two field campaigns in Galveston Bay (Fig. S1). The same will be discussed briefly in "Results" and "Discussion" Sections. In a separate study, we are presenting a more detailed analysis of CDOM and particulate absorption dynamics and fluxes associated with Hurricane Harvey.*

"Page 6 SPM uncertainty? Sensor used for bbp measurements and data processing/corrections?"

Response: *We did not have the replicates of SPM measurements, therefore uncertainty in sample collection and filtering process is unknown. However, the uncertainty in weighing scale measurement is $\pm 0.1$ mg. We used the WETLabs VSF-3 (470 nm, 530 nm, and 660 nm) and ECO BB (532 nm) backscattering sensors for $b_{bp}$ measurements (D'Sa et al., 2006).*

"Eq. 2: Rrs (residual)? Please define and justify"

Response: *Rrs (residual) is attributed to residual sky-radiance. Generally, it is assumed that water-leaving radiance is zero at 750 nm in open ocean, which may not hold well in coastal and estuarine waters. We moved residual wavelength to 950 nm in this study with an assumption that water-leaving radiance is zero at 950 nm (Which may not be true in highly turbid regions of Galveston Bay such as near the Trinity River). We have updated text accordingly in the revised manuscript.*

"Eqs. 1-3: provide physical unit for each parameter"

Response: *Units will be provided in revised manuscript.*

[Figure]

**Figure S1: Application of sensor-specific QAA tuning to obtain the maps of $a_{tnw}443$ using a) VIIRS, b) MODIS-Aqua and c) Sentinel3 OLCI on October 29, 2017, and d) VIIRS and e) Landsat8 OLI on September 30, 2017 and September 29, 2017, respectively. The validation of these maps with field observation along the transect (St. 1 to St. 14) is shown in (I) for figs. 8a-8c and in (II) for figs. 8d & 8e. The maps of $b_{btnw}470$ were obtain similarly for f) VIIRS (October 29, 2017), g) MODIS-Aqua, h) Sentinel3 MSI, i) VIIRS (September 30, 2017), and j) Landsat8 OLI (September 29, 2017) with their validation results in (III) and (IV), respectively. Parameter values beyond the upper limit of tuned QAA ($\rho > 0.65$) is masked in white.**

Reference:

D'Sa, E. J., Miller R. L., and Del Castillo C.: Bio-optical properties and ocean color algorithms for coastal waters influenced by the Mississippi River during a cold front, Applied Optics, 45, 7410–7428, 2006

---

## Author Response (AR1)

Response: *We have added calibration coefficients for other ocean color (e.g., MERIS, MODIS-A, Sentinel3 OLCI, and SeaWiFS) and land observing (e.g., Landsat8 OLI and Sentinel2 MSI) sensors in revised manuscript (Table S1). Additionally, we also showed the performance of proposed calibration coefficients through performance comparisons between the sensors and field observations (Figure S1). We added Figure 10, Table 3, new sub-sections in "Result" and "Discussion". Based on the application of this study to a large number of ocean color sensors, we will modify the title to "An estuarine tuned Quasi-Analytical Algorithm (QAA-V): assessment and application to satellite estimates of SPM in Galveston Bay following Hurricane Harvey"*

*Result Section*
*3.6 Extending the QAA-V tuning to various satellite sensors*
*The estuarine-specific green to red band tuning was further applied to evaluate and to extend its applicability to past and present ocean color (e.g., SeaWiFS, MERIS, MODIS-Aqua, and Sentinel3 OLCI) and land-observing sensors (Landsat8 OLI and Sentinel2 MSI) (Table 3). The validation analysis showed promising performance of QAA tuning in obtaining total non-water absorption coefficient ($a_{tnw}443$) and total-non water backscattering coefficient ($b_{btnw}470$) in optically complex and shallow waters of Galveston Bay (Fig. 10). Overall, different satellite sensors showed similar trends of $a_{tnw}443$ and $b_{btnw}470$ along the transect despite having different spectral and spatial sensor resolutions (Fig. 10I – 10IV). The MREs were ~15 %, 9 %, and 12 % for $a_{tnw}443$ retrievals from VIIRS, MODIS-A, and Sentinel3 OLCI sensors, respectively (Fig. 10a-c & 10I), whereas they were ~26 %, 7 %, 22 % for $b_{btnw}470$ retrievals on October 29, 2017 (Fig. 10f-h & 10III). For Landsat8 OLI, these MRE were ~20 % and ~10 % for $a_{tnw}443$ and $b_{btnw}470$, respectively on September 29, 2017 (Fig. 10e, 10j, 10II, & 10IV).*

*Discussion Section*
*4.4 Application of the QAA tuning to various ocean color and land observing sensors*

*Sensor-specific QAA tuning (e.g., QAA-V for VIIRS, MODIS, Landsat-8 OLI, and Sentinel3 OLCI) showed overall valid retrieval of absorption and backscattering coefficients with various ocean color and land-observation sensors (Fig. 10). Although satellite-derived values and trends of $a_{tnw}443$ and $b_{btnw}470$ are similar to the field observations, the observed discrepancies could have in addition to the uncertainties in field measurements due to several sources of errors. For example, it is well-known that satellite products suffer from large errors in the blue region especially in coastal waters due to atmospheric correction (Table 4; Supplementary S4). The large errors between field and Landsat8 OLI derived $a_{tnw}443$ could have resulted due to the fact that the QAA processing chain uses these erroneous blue Rrs values to obtain $a_{tnw}$ at the blue wavelengths (Table 2; Level 3). However, the $b_{btnw}$ retrievals at the blue wavelengths are unaffected by the blue Rrs inputs (Table; Level 2). Likewise, the errors were relatively smaller at the reference wavelength (Table 4) because the proposed QAA tuning avoided using blue wavelengths in the primary step of getting $a_{tnw}$ and $b_{btnw}$ at a reference wavelength. Hence, the atmospheric correction procedure is an important step that would impact the performance of QAA-V in ocean color applications of shallow estuarine and near-shore waters.*

*Another important discrepancy among various sensor maps is the number of masked pixels in shelf waters (Fig. 10). We have used a VIIRS band-ratio based threshold ($\rho=0.65$) to separate green waters (e.g., productive coastal waters) and blue waters (e.g., open ocean) (Eq. 15) and applied it to various satellite sensors. Although this threshold worked well for the sensors with similar green and red bands (e.g., MODIS-A and VIIRS), it did not perform as well for Sentinel 3 OLCI and Landsat 8 OLI because of notable differences between the green and red bands of these sensors and the VIIRS sensor (Table 3). Although, the proposed threshold works well to represent estuarine and near-shore waters for various ocean color and land-observing sensors, it could be optimized for each satellite sensor.*

"Moreover, authors do not motivate the choice of this sensor. For example, why choose VIIRS while Landsat-8/OLI or Sentinel-2/MSI provide data with a better spatial resolution (which is crucial for coastal applications)? I recommend to the authors to make explicit the choice of VIIRS. I also recommend that authors provide in a table the calibration coefficients for other ocean color sensors."

Response: *We focused on VIIRS primarily due to our earlier work based on this sensor (Joshi et al. 2017). We appreciate the reviewer's suggestion for using land observing sensors for coastal applications. We worked on several ocean color and land observing sensors and provided their calibration coefficients and validation analysis in revised manuscript (Table S1 and Fig. S1). We strongly feel that our study is a wider ocean color application than for the VIIRS sensor. We included the following lines in "2.3 QAA-V processing chain in Materials and methods" to initiate discussion on this additional work,*

*To extend and to evaluate the applicability of estuarine-specific QAA tuning, it was further applied to various ocean color (Sentinel3 OLCI, MODIS-Aqua, MERIS, and SeaWiFS) and land observation sensors (Landsat8 OLI and Sentinel2 MSI). The calibration coefficients for obtaining total non-water absorption coefficient at a reference wavelength ($a_{tnw}$ ($\lambda_0$); Level 1B in Table 2) are given in Table 3.*

"2- Authors mention that the QAA-V can be applied in optically shallow waters. For instance, p.1, lines 8-11 : "The standard quasi-analytical algorithm (Lee et al., 2002) was tuned as QAA-V using a suite of synthetic data and in-situ measurements to improve its performance in OPTICALLY complex and shallow estuarine waters". p.4, lines 5-7 : "In this study, we present a tuned multiband Quasi-Analytical Algorithm (QAA-V) optimized to estimate IOPs in

OPTICALLY shallow and near-shore waters for the Visible and Infrared Imaging Radiometric Suite (VIIRS) ocean color sensor". or, p.19, lines 12-14: "The QAA-V may not perform satisfactorily in optically deep waters as the empirical relationships were designed specifically for the optically shallow environments". I think this error comes from a lack of knowledge of the authors of the definition of "optically shallow waters". "Optically shallow waters" doesn't mean "shallow waters". A definition can be found in the IOCCG Report Number 3 (2000). "Optically shallow implies that the product of the diffuse attenuation coefficient Kd and the geometric depth z is small" (p.33). "Coastal waters can also be optically shallow, so that water-leaving radiance is affected by bottom reflectance" (p.94). "Where coastal waters are optically shallow, algorithms for water-column constituents need to remove contributions from bottom reflectance" (p.99). For highly absorbing and turbid waters (which is the case of this study), we can expect a high value of Kd and consequently a high value of the product of Kd and z (even in the case where z is small). It is difficult to believe that the water-leaving radiance is significantly affected by bottom reflectance. More important, the QAA is not designed to take into account the contribution from bottom reflectance. No study has ever shown that QAA works in optically shallow waters. By consequence, I recommend to the authors to replace "optically shallow waters" by "shallow waters. Moreover, for clarity, the author should also specify that the QAA-V was developed for optically deep waters."

Response: *Thank you for pointing this out. We appreciate the reviewer's detailed comment and suggestion for this error. We have replaced "optically shallow waters" by "optically complex and shallow waters" or "shallow waters" at several locations in manuscript.*

"3- The results do not really demonstrate the interest of using QAA rather than existing algorithms (for instance, Nechad et al. (2010) or Han et al.(2016)) to estimate SPM. P.16, lines 15-22, authors mention the limits of the use of Rrs to estimate SPM before to underline the interest of the use of bbp. They forget to mention the strong limits of this alternative method. bbp is not directly measured. The inversion model used to derivate bbp generates an inherent error that propagates for the SPM inversion. Another source of error is due to the fact that the bbp to SPM ratio is not constant and its value depends of the particle nature. I recommend to authors to discuss precisely the limits of the "bbp method" for the SPM estimation."

Response: *Thank you very much for this suggestion. Because we obtained reasonable estimates of backscattering coefficients from QAA-V processing chain, we decided to include in latter part of this study (a case study of post-hurricane SPM dynamics in Galveston Bay) as an application of QAA-V for obtaining Level-2 products such as SPM (this study). We agree that there are a few limitations for bbp-based approach because the uncertainty in satellite estimates of bbp (and thus SPM estimates). We listed these limitations in Section 4.2 according to the reviewer's suggestion.*

*Some factors that limit the efficacy of "bbp method" and cause differences between field and satellite SPM matchups include:*
*1) propagation from various steps of the QAA-V processing chain to $b_{btnw}532$ (e.g., 20–30 %, Fig. 7c) and hence, further down to the SPM inversion, 2) the uncertainty in the atmospheric-corrected green and red Rrs (e.g., 5–20 %, Table 4), 3) the uncertainty in SPM–$b_{btnw}532$ relationship due to limited observations, 4) the assumption of linearity beyond in SPM–$b_{btnw}532$ model beyond the instrument threshold which may not hold well because $b_{btnw}532$ to SPM ratio depends on the particle nature and it may not be always constant especially in highly turbid waters, and 5) errors in SPM measurements.*

**Response to Anonymous Reviewer #2**

"This study presents a tuning of the well-known standard quasi-analytical algorithm (QAA, Lee et al., 2002) to improve its performance in optically complex and shallow estuarine waters. This tuning is based on both synthetic data and in-situ measurements. The improvement due the tuned algorithm (called QAA-V) is assessed based on once again synthetic data (Hydrolight computations) and in-situ measurements in several US coastal and estuarine waters. The performance of the tuned algorithm is proved to be reasonable so as the improvement when compared to QAA last version (v6). Finally, QAA-V is applied to VIIRS satellite data recorded over turbid coastal waters (Galveston Bay, Texas) to retrieve then map the particulate light backscattering coefficient at 532 (bbp532) actually converted into suspended particulate matter concentrations (SPM) using an empirical relationship. SPM maps generated after an intense flooding event are analyzed using river discharge and wind speed/direction data. Overall, the manuscript presents a significant piece of work with interesting results. Moreover, it is well organized and written. The most convincing part is certainly the tuning of the QAA algorithm for shallow estuarine waters and the efforts made to validate the results obtained."

Response: *We appreciate that the reviewer found this work interesting and significant.*

"The less convincing section is the retrieval and mapping of SPM concentrations in Galveston Bay using VIIRS satellite data as: QAA allows retrieving the light absorption and particulate coefficients of colored water constituents on top of pure water (anw and bbnw, respectively), i.e. much more than SPM concentration. "One would expect maps of anw and bbnw to be presented and analyzed at the end of the study that could highlight interesting spatial and temporal variations of CDOM and SPM composition and size distribution over the study area. My recommendation is to accept the manuscript for publication if the authors can address in more details this last comment, at least in the Discussion and Conclusions sections."

Response: *Because we obtained reasonable estimates of backscattering coefficients from QAA-V processing chain, we decided to include in latter part of this study an application of QAA-V for investigating post-hurricane SPM dynamics in Galveston Bay. In the revised version, we showed results of $a_{tnw}$ and $b_{btnw}$ maps for VIIRS and additional ocean color and land observing sensors during our field campaigns in Galveston Bay (Table S1, Fig. S1). The same are discussed briefly in "Results" and "Discussion" Sections. In a separate study, we are presenting more detailed study of CDOM and particulate absorption dynamics and fluxes associated with Hurricane Harvey.*

"Page 6 SPM uncertainty? Sensor used for bbp measurements and data processing/corrections?"

Response: *We did not have the replicates of SPM measurements, therefore uncertainty in sample collection and filtering process is unknown. However, the uncertainty in weighing scale measurement is $\pm 0.1$ mg. We have used WETLabs VSF-3 (470 nm, 530 nm, and 660 nm) and ECO-BB (532 nm) sensors for $b_{bp}$ measurements (D'Sa et al., 2006).*

"Eq. 2: Rrs (residual)? Please define and justify"

Response: *Rrs (residual) is attributed to residual sky-radiance. Generally, it is assumed that water-leaving radiance is zero at 750 nm in open ocean, which may not hold well in coastal and estuarine waters. We moved residual wavelength to 950 nm in this study with an assumption that water-leaving radiance is zero at 950 nm (Which may not be true in highly turbid regions of Galveston Bay such as near the Trinity River). We have updated text accordingly in revised manuscript.*

"Eqs. 1-3: provide physical unit for each parameter"

Response: *Units will be provided in revised manuscript.*

**Table S1:  The calibration coefficients for sensor-specific QAA tuning. $\lambda_0$ is a sensor-specific reference wavelength.**

| Sensor | $\rho = \log_{10}\left(\dfrac{R_{rs}{}^{0-}(\lambda_0)}{R_{rs}{}^{0-}(\lambda_1)}\right)$ | $a_{tnw}(\lambda_0) = 10^{(a+b\times\rho+c\times\rho^2)}$ (Level 1B —Table 2) | | | | | |
|---|---|---|---|---|---|---|---|
| | | $\rho < 0.25$ | | | $\rho \geq 0.25$ and $\rho \leq 0.65$ | | |
| | | a | b | c | a | b | c |
| VIIRS | $\lambda_0 = 551$ nm & $\lambda_1 = 671$ nm | 0.139 | -1.788 | 0.490 | 0.406 | -2.940 | 0.928 |
| MODIS-Aqua | $\lambda_0 = 555$ nm & $\lambda_1 = 667$ nm | 0.091 | -1.800 | 0.560 | 0.275 | -2.674 | 0.813 |
| Sentinel3 OLCI | $\lambda_0 = 560$ nm & $\lambda_1 = 674$ nm | 0.176 | -1.830 | 0.528 | 0.397 | 2.940 | 0.800 |
| MERIS | $\lambda_0 = 560$ nm & $\lambda_1 = 665$ nm | 0.081 | -1.868 | 0.688 | 0.314 | -2.733 | 0.713 |
| SeaWiFS | $\lambda_0 = 555$ nm & $\lambda_1 = 670$ nm | 0.128 | -1.792 | 0.505 | 0.276 | -2.742 | 0.842 |
| Sentinel2 MSI | $\lambda_0 = 560$ nm (Band 3) & $\lambda_1 = 665$ nm (Band 4) | 0.0814 | -1.868 | 0.688 | 0.223 | -2.732 | 0.740 |
| Landsat8 OLI | $\lambda_0 = 560$ nm (Band 3) & $\lambda_1 = 655$ nm (Band 4) | -0.087 | -1.900 | 0.952 | 0.057 | -2.667 | 0.753 |

[Figure]

**Figure S1: Application of sensor-specific QAA tuning to obtain the maps of $a_{tnw}443$ using a) VIIRS, b) MODIS-Aqua and c) Sentinel3 OLCI on October 29, 2017, and d) VIIRS and e) Landsat8 OLI on September 30, 2017 and September 29, 2017, respectively. The validation of these maps with field observation along the transect (St. 1 to St. 14) is shown in (I) for figs. 8a-8c and in (II) for figs. 8d & 8e. The maps of $b_{btnw}470$ were obtain similarly for f) VIIRS (October 29, 2017), g) MODIS-Aqua, h) Sentinel3 MSI, i) VIIRS (September 30, 2017), and j) Landsat8 OLI (September 29, 2017) with their validation results in (III) and (IV), respectively. Parameter values beyond the upper limit of tuned QAA ($\rho > 0.65$) are shown masked in white.**

Additional references:

Ishan D. Joshi, Eurico J. D'Sa, Christopher L. Osburn, Thomas S. Bianchi, Dong S. Ko, Diana Oviedo-Vargas, Ana R. Arellano, Nicholas D. Ward, 
[revised manuscript text omitted]